psychology/cognition

first impressions, facial traits, other-race effect, racism, inter-group bias

**Author for correspondence:**
Richard Cook
e-mail: richard.cook@bbk.ac.uk

# Why is the literature on first impressions so focused on White faces?

Richard Cook[1] and Harriet Over[2]

[1]Department of Psychological Sciences, Birkbeck, University of London, Malet Street, London WC1E 7HX, UK
[2]Department of Psychology, University of York, Heslington, York YO10 5DD, UK

RC, 0000-0003-2370-3086; HO, 0000-0001-9461-043X

We spontaneously attribute to strangers a wide variety of character traits based on their facial appearance. While these first impressions have little or no basis in reality, they exert a strong influence over our behaviour. Cognitive scientists have revealed a great deal about first impressions from faces including their factor structure, the cues on which they are based, the neurocognitive mechanisms responsible, and their developmental trajectory. In this field, authors frequently strive to remove as much ethnic variability from stimulus sets as possible. Typically, this convention means that participants are asked to judge the likely traits of White faces only. In the present article, we consider four possible reasons for the lack of facial diversity in this literature and find that it is unjustified. Next, we illustrate how the focus on White faces has undermined scientific efforts to understand first impressions from faces and argue that it reinforces socially regressive ideas about 'race' and status. We go on to articulate our concern that opportunities may be lost to leverage the knowledge derived from the study of first impressions against the dire consequences of prejudice and discrimination. Finally, we highlight some promising developments in the field.

## 1. Introduction

When we first encounter a stranger, we spontaneously attribute to them a wide variety of character traits (e.g. intelligence, trustworthiness, dominance) based on their facial appearance [1–3]. While these first impressions often have little or no basis in reality, they exert a strong influence over our behaviour [4]. In economic games, adults invest more resources in individuals who appear trustworthy than in individuals who appear untrustworthy [5]. First impressions may also affect legal judgements, criminal sentencing [6–8] and even the outcome of elections [9–11]. Repeated instances of so-called 'faceism' may resemble systematic prejudice against individuals with untrustworthy facial appearance [2,9,12].

**Figure 1.** The kinds of stimuli used in the first impressions literature. (*a*) Examples of the synthetic facial stimuli generated by the computer model described by Oosterhof & Todorov [13]. (*b*) Examples of adult face stimuli taken from the set of Karolinska Directed Emotional Faces [28] (https://www.kdef.se/). (*c*) Examples of naturalistic 'ambient' images supplied by the authors with the permission of the individuals depicted.

Cognitive scientists have revealed a great deal about these first impressions, including the factor structure of trait judgements and the cues on which they are based [13–15]. Participants generate consistent impressions of faces even when exposed to them for less than 100 ms [16,17]. The trait judgements made by children resemble those of adults by the age of five, if not earlier [18–20]. In terms of neural substrate, findings from neuroimaging suggest the amygdala plays an important role, particularly in inferences of trustworthiness [21–24].

Researchers typically present a facial image and ask participants to judge the likely traits of the person depicted. In some studies, the images depict synthetic faces generated by a computer model [5,13,25–27] (figure 1*a*). In other studies, photographs of real human faces are used. Some authors prefer to use tightly controlled stimuli, where the people depicted are photographed under consistent lighting conditions, while exhibiting similar facial expressions [15,17,29] (figure 1*b*). Other authors prefer to use naturalistic 'ambient' images that vary widely in pose, expression and lighting conditions [14,30–34] (figure 1*c*).

In this field, authors typically use stimulus sets that comprise only White faces, often with little or no explanation (e.g. [2,5,12,14–17,19–23,25–27,29,31–42]). This is by no means a historical problem; for example, a number of studies published within the last few years maintain this convention (e.g. [12,19,23,27,31,32,34–36,41,42]). In the present article, we consider four possible reasons for the lack of facial diversity in this literature and find it unjustified. We go on to argue that the lack of facial diversity has undermined scientific efforts to understand first impressions from faces and potentially reinforces socially regressive ideas about 'race' and status. Following this, we consider whether opportunities are being lost to use knowledge derived from studies of first impressions to mitigate prejudice and discrimination. Finally, we highlight some promising developments in the field.

## 2. Two parallel literatures

The literature on first impressions of faces has sought to understand the trait inferences made about individuals. For example, why two White people with similar facial appearance are judged differently because one has narrower eyes than the other [2,3,43]. This literature has strong roots in vision

science, cognitive psychology and cognitive neuroscience. A second closely related field, one with strong roots in social psychology, has sought to understand inter-group biases [44–47].

Importantly, the literature on inter-group bias includes overwhelming evidence that perceived ethnic groupings are associated with particular stereotypes [46–48]. For example, White American participants often associate Black Americans with negative character traits including stupidity, laziness and aggression [47,49]. Similarly, White American participants may assume that individuals of East Asian appearance are intelligent and hardworking, while people with Jewish appearance are cold and calculating [47,48]. The influence of these stereotypes is devastating, contributing to systematic discrimination against particular groups [46,50]. There is considerable evidence that racial stereotypes have a cultural origin [49,51–53].

As described above, the literature on first impressions from faces has focused overwhelmingly on the trait inferences made about White faces. However, where authors have employed faces of colour, results suggest that cues to perceived ethnicity exert a strong influence on trait judgements [54–58]. For example, Stanley et al. [54] found that individuals who exhibited a pro-White implicit bias were likely to judge White faces as more trustworthy than Black faces. Similarly, Zebrowitz et al. [55] found that White American participants judged Black faces to be more dominant than White or Korean faces, while Black American and Korean participants judged Black faces to be less dominant than White or Korean faces.

We believe that a comprehensive account of first impressions from faces needs to integrate these findings about racial stereotypes. It must explain not only why White people with narrow eyes are judged to be untrustworthy but also why Black people are often judged to be untrustworthy and aggressive. All too frequently, however, authors seek to make their stimulus sets as ethnically homogeneous as possible and dismiss the effects of racist stereotypes as a source of noise [14,31,33,34] or a potential confound—a qualitatively different phenomenon the influence of which needs to be controlled for [2,15,16,38,40].

# 3. Why is there a lack of facial diversity in the first impressions literature?

It is not always clear why faces of colour are excluded from stimulus sets. Many studies in this field use only White faces without any justification [5,15,17,19–23,25,32,35–37,41]. Indeed, we have been guilty of this in our own research [19,37,42]. Below, we consider four possible reasons why authors may have used ethnically homogeneous stimuli.

## 3.1. Does the use of diverse faces introduce a confound?

One potential justification for using ethnically homogeneous stimuli is that the inference of traits from cues to ethnicity (e.g. skin colour) and from other facial features (e.g. square jaw, narrow eyes) are qualitatively different phenomena [2,15,16,38,40]. Some authors have alluded to this possibility without further elaboration [15,38,40]. In a notable exception, however, Todorov et al. [2] argued that faceism has a perceptual basis insofar as there is a tight coupling between variation in the appearance cues presented and the different traits inferred by participants. By contrast, inter-group effects are deemed to have a non-perceptual basis. It is argued that the indiscriminate application of group stereotypes produces a relatively weak relationship between facial appearance and the traits inferred.

This rationale does not withstand scrutiny. First, it is not the case that participants indiscriminately apply the same trait profile to all individuals from a particular group. For example, among people who identify as Black, individuals with darker skin experience greater discrimination in many contexts [59–61]. People with a more stereotypically Black appearance receive harsher sentencing decisions [61]. Similarly, subtle differences in feature shape influence the affective reactions of White participants to Black faces [62]. These findings reveal sensitivity to the particular facial appearance of individuals of other perceived ethnicities.

Second, the inference of traits from cues to ethnicity seems closely comparable to the inference of traits from sexually dimorphic cues, thought to be a key *perceptual* determinant of facial trait evaluation [2]. In both cases, facial cues can be used to categorize the person depicted (e.g. as Black or White or as male or female) and particular attributions may arise from the activation of group stereotypes. For example, Black individuals may be judged to be aggressive and females may be judged to be caring and submissive [47,48]. Like facial cues to ethnicity, facial cues to sex include skin

tone and feature shape. Why then treat sexually dimorphic cues as a key perceptual determinant of facial trait evaluation, but cues to ethnicity as a confound?

This example reveals that the focus on White faces does not remove the influence of stereotypes. Indeed, a White face might activate a host of group stereotypes. For example, when a White face with stereotypically Jewish appearance is presented, some participants are likely to attribute traits consistent with an anti-Semitic group stereotype [47,48]. Similarly, male faces with effeminate features may engage homophobic stereotypes and be judged extroverted or submissive. We also note that certain types of character (e.g. heroes and villains, jocks and geeks) are also prone to stereotypical depiction in film, TV, comics and storybooks. The activation of these character stereotypes may afford a range of attributions about courage, trustworthiness, and academic and sporting ability. Far from being a confound arising from a different phenomenon, group and character stereotypes are likely to be fundamental components of first impressions from faces. These effects, together with those effects driven by the (mis)perception of facial emotion [63,64], likely account for the majority of consensus impressions.

## 3.2. Do participants lack the perceptual expertise necessary to evaluate diverse faces?

A second potential justification for using ethnically homogeneous stimuli is that it mitigates the influence of the so-called 'other-race' effect [14,27,31,33,34]. The other-race effect refers to a phenomenon whereby some individuals are better at distinguishing and identifying individuals from their own ethnicity [65–68]. In the light of the other-race effect, it is possible that participants lack the necessary perceptual expertise to produce meaningful ratings of diverse faces; thus, White individuals should only be asked for their first impressions of White faces. Once again, the rationale behind this justification does not withstand scrutiny.

Where this rationale is cited, authors do not explain why subtle differences in face perception ability should substantially distort the resulting patterns of first impressions. The inference of traits from appearance appears to depend on a relatively crude facial analysis. Individuals with severe socially debilitating face recognition problems—those with developmental prosopagnosia [69,70]—make broadly typical judgements of facial traits [71]. This condition impairs the perceptual encoding of face shape [72], disrupts the interpretation of facial emotion [73] and is associated with imprecise classification of facial sex [74]. Compared to the severe face recognition problems seen in developmental prosopagnosia, deficits associated with the other-race effect are relatively mild [75].

Moreover, not everyone shows other-race effects. People are thought to develop expertise for the types of faces to which they are exposed [65,68,76]. For example, adults of Korean origin adopted by White families living in France showed better recognition of White faces than of Asian faces [76]. Where first impressions research is conducted in diverse urban centres, it seems unjustified to assume that local participants lack perceptual expertise for diverse faces. Individuals growing up in London, Paris or New York will frequently have to identify individuals (e.g. teachers, classmates, co-workers) from a range of ethnic backgrounds. It is this 'individuation experience' that is thought to be crucial for the development of perceptual expertise for faces [77,78]. Note, it is easy enough to identify (and exclude) participants who are unable to provide reliable first impressions; for example, faces can be rated twice and the consistency of participants' ratings examined.

As we explain above, authors frequently (typically) offer no explanation for the lack of facial diversity in stimulus sets. However, certain aspects of this literature suggest that concerns about raters' face perception ability are not widespread among researchers or, at least, manifest inconsistently. In some first impressions studies, people of colour are unable to participate as raters [15,27,34,38]. In other studies, people of colour are able to participate as raters [19,29,32,41]. More typically, however, authors provide little or no description of raters' background or ethnicity [5,13,16,17,20–22,31,33,35,36,39]. This suggests that the other-race effect is not a major concern for many authors and reviewers.

Similarly, little or no attempt has been made to control for the other-age effect, a bias similar to the other-race effect whereby participants show superior individuation of own-age faces [79]. In several studies, participants in their late teens have been asked to evaluate ambient images of people 50 or 60 years older [14,31,38]. Authors appear to accept that these participants possess sufficient perceptual expertise to judge the character traits of older faces without question.

Finally, much research in this area incorporates unrealistic synthetic White faces that caricature cues to particular traits [13]. It is unclear to what extent perceptual expertise developed with real faces aids the

individuation of such unusual stimuli [80]. Given that synthetic faces with which participants have little or no perceptual experience can be valuable stimuli, then why not faces of colour?

## 3.3. Might evaluations based on ethnicity overshadow other types of inference?

It is possible that some researchers view first impressions from cues to ethnicity and first impressions from other types of structural face cue as qualitatively similar, but fear that the former may overshadow the latter. When diverse faces are presented interleaved, differences in face ethnicity are potentially rendered highly salient [54,81,82]. Moreover, cues to ethnicity likely exert a strong influence over the attribution of particular traits (e.g. intelligence [47]). Consequently, more subtle effects might be harder to detect (e.g. the influence of eyelid openness and mouth curvature on attributions of intelligence [15]). Participants may attend less to subtle facial differences and/or weight this variation less when making trait evaluations.

When addressing certain questions, we can understand concerns about overshadowing (e.g. when seeking to reveal the influence of a subtle facial cue on a particular trait evaluation). In many cases, however, concerns about overshadowing seem unjustified. Perceived ethnicity remains one of the most influential cues for social evaluations outside the laboratory [83–85]. If one accepts that trait inferences based on cues to ethnicity are qualitatively similar to other types of first impression, then it follows that cues to ethnicity are a major component of the phenomenon researchers are trying to understand. It, therefore, seems essential to incorporate diverse faces in any study seeking to reveal the neurocognitive mechanisms that mediate trait evaluations [17,21,22], in data-driven attempts to determine which cues determine first impressions [13,14] and in studies designed to assess the validity and consequences of first impressions [4,5,86,87].

Moreover, we see no reason why concerns about overshadowing should mean faces of colour are entirely omitted from studies of first impressions. For example, the use of all-Black or all-Asian faces would control equally well for differences in ethnicity. Alternatively, authors could examine the effect of a particular cue (e.g. mouth curvature) using blocked or between-subjects designs, in which participants encounter different facial ethnicities (e.g. White, Black, Asian) in different blocks or different conditions. This approach would allow researchers to determine whether the influence of a particular cue varies as a function of face ethnicity. Note, the use of blocked or between-subjects designs also addresses any potential concern about the influence of low-level image variation (e.g. luminance differences) that may arise when diverse faces are interleaved within trial blocks.

It may well be the case that increasing the diversity present within stimulus sets changes the way people make trait evaluations (e.g. some features may be rendered more salient, while other features may appear less salient). Importantly, however, this approximates more closely the situation that participants encounter outside the laboratory. Western societies are increasingly diverse. The faces we encounter on the train to work or in our local coffee shop are diverse, and our spontaneous trait judgements are made in these diverse contexts. If the influence of a particular cue is not seen when diverse faces are interleaved, this raises the question: how much influence does that cue have outside the laboratory?

Once again, we note that authors typically offer little or no justification for their use of ethnically homogeneous White face stimuli. It is, therefore, hard to gauge whether concerns about overshadowing by cues to ethnicity are widespread. However, there is reason to believe that many authors are unconcerned about overshadowing effects *per se*. In just the same way that interleaving ethnically diverse faces may render cues to ethnicity salient, interleaving male and female faces may also render sexually dimorphic cues salient. Like racial stereotypes, gender stereotypes are also strongly associated with expectations about intelligence, competence, trustworthiness and dominance [47,48]. Thus, both sexually dimorphic cues and cues to ethnicity might be expected to overshadow more subtle face-trait mappings. Nevertheless, the practice of interleaving male and female faces is commonplace in the first impressions literature (e.g. [13–15,17,36]). Similarly, studies that employ ambient images interleave target faces that vary widely in age and emotional expressions [14,30–34]. Again, there seems little concern that this variability might overshadow trait inferences based on more subtle cues.

## 3.4. Are there logistical impediments that prevent the use of diverse stimuli?

Historically, some authors may have used White facial stimuli out of convenience. Developing a bespoke set of stimulus images, and obtaining the necessary usage rights and pre-ratings, is not a trivial

undertaking. Consequently, many authors use stimuli from pre-existing collections of face images that are freely available for academic use. Several older collections comprise mostly or exclusively White faces, including The Radboud Faces Database [88], The Karolinska directed emotional faces [28], The Glasgow Unfamiliar Face Database [89] and The Dartmouth Database of Children's Faces [90]. Quick and easy access to these resources may have encouraged the use of all-White faces in first impressions research.

More recently, diverse collections of facial stimuli have been made available to researchers. These new image sets include the Bogazici Face Database [91], the Chicago Face Database [92], the RADIATE Stimulus Set [93], the American Multiracial Faces Database [94] and the MR2 Face Database [95]. Access to these resources means that there are no longer logistical impediments preventing the use of diverse face stimuli in first impressions research.

We note, the lack of diverse image sets—historically—does not explain the use of all-White sets of computer-generated faces [5,13,25,26,35]. FaceGen Modeller (Singular Inversions Inc.), the programme most often used to generate stimulus faces (figure 1a), permits the generation of Black and Asian faces without any additional difficulties (e.g. [96]). Nor does it explain the use of all-White sets of ambient images [14,33,34,38]. These image sets are not sourced from publicly available databases; rather, they are obtained by the authors from various websites identified via internet searches.

# 4. Consequences of excluding faces of colour

## 4.1. Scientific consequences

Thus far, we have argued that the use of ethnically homogeneous stimuli in first impressions research is unjustified. Next, we argue that the failure to include faces of colour in stimulus sets has had detrimental consequences for scientific enquiry into first impressions. We illustrate these deleterious effects with reference to three questions that have been integral to the study of first impressions. We offer these as illustrative examples rather than as an exhaustive account of the deleterious consequences of focusing primarily on White faces.

### 4.1.1. How accurate are first impressions?

There is much interest in the veracity of first impressions from faces [4,87,97]. Some researchers maintain that first impressions contain at least a 'kernel of truth' [86,87]. Consistent with this claim, previous research has shown that White individuals who appear trustworthy to raters are, in fact, more generous in a range of behavioural tasks than are White individuals who appear untrustworthy [86]. This claim has proved controversial, however. Other researchers maintain that the majority of first impressions are inaccurate [4,97]. Incorporating more diverse faces into stimulus sets would reveal the extent of inaccuracy in first impressions. There is abundant evidence that negative racial stereotypes are systematically inaccurate [98,99].

### 4.1.2. How much consensus is there in first impressions?

There is also much interest in the degree of consensus shown between participants in their trait judgements. Previous research suggests that there are high levels of inter-rater agreement in who appears trustworthy and untrustworthy [2,13,17]. However, the lack of diversity in stimulus sets, combined with majority White participant samples, likely artificially inflates levels of inter-rater agreement [100]. Attitudes towards perceived racial groups [45,101] and ethnic facial stereotypes [102,103] vary widely within and across cultures (see also [54,55,58]). Greater diversity is, therefore, likely to reveal more heterogenous first impressions than is currently appreciated in the literature.

### 4.1.3. Where do first impressions of faces come from?

In recent years, we have argued that first impressions are the products of culturally acquired mappings between visual representations of different face shapes and representations of the various trait profiles that other people may possess [19,42,43,100,104,105] (figure 2). According to this account, trait inferences from cues to ethnicity are closely comparable to trait inferences from other facial cues. An alternative perspective is that some first impressions have an innate basis; that the mechanism or knowledge responsible is an evolutionary adaptation for identifying trustworthy collaborators and good leaders [30,106–109].

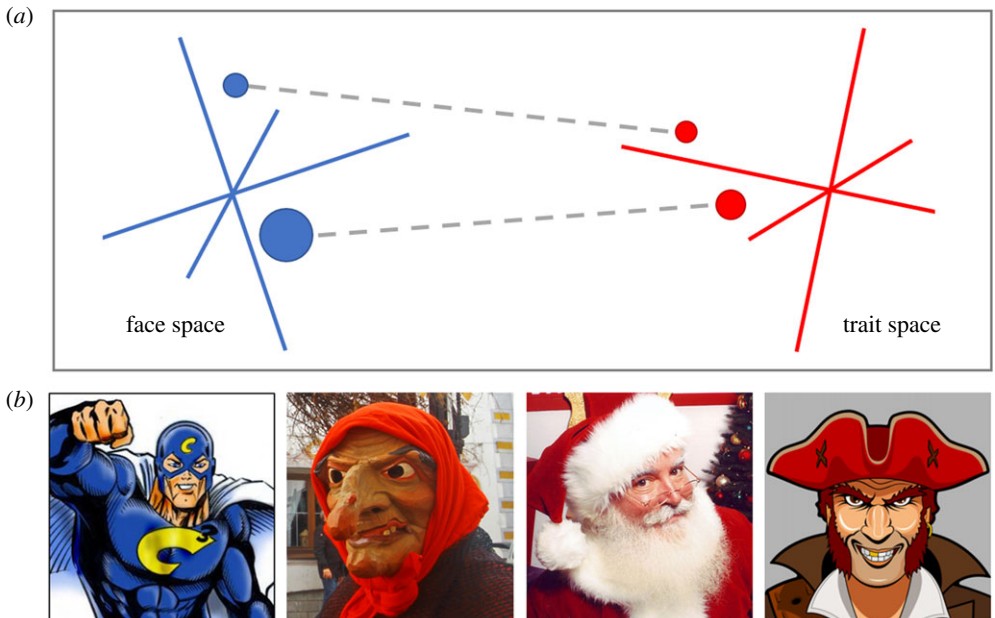

**Figure 2.** The trait inference mapping model (TIM) of first impressions [43]. (*a*) According to TIM, first impressions are the product of learned mappings that allow excitation to spread from perceptual representations of face shape (points in face space) to representations of trait profiles that others may possess (points in trait space). (*b*) The mappings responsible for consensus impressions are acquired culturally following exposure to stereotypical depictions of certain character types (e.g. 'good guys' and 'bad guys') in story books, film, TV, ritual and propaganda. The images in (*b*) were sourced through https://commons.wikimedia.org/. The first panel is cropped from an illustration of a superhero (artwork by Mitch Hallock; CC-BY-SA 3.0 (https://creativecommons.org/licenses/by-sa/3.0/)). The second panel is cropped from a photograph of a person in a witch costume (photograph by Iryna Pustynnikova; CC BY-SO 4.0 (https://creativecommons.org/licenses/by-sa/4.0/)). The third panel is cropped from a photograph of Jonathan Meath dressed as Santa Claus (photography by Jonathan Meath; CC BY-SA 2.5 (https://creativecommons.org/licenses/by-sa/2.5/)). The fourth panel is cropped from an illustration of a pirate (artwork by Cyrille R W Chaussepied; CC BY-SA 4.0 (https://creativecommons.org/licenses/by-sa/4.0/)).

The foregoing work on the accuracy and consistency of first impressions informs this origins debate. The ability to accurately infer the traits of others from their facial appearance would likely convey a competitive advantage upon our ancestors [106]. If our first impressions have some veracity, this would, therefore, suggest a possible route by which the mechanism responsible might become a genetic adaptation [106]. Similarly, high levels of consensus, where observed, have been cited as evidence for 'universality' and an innate explanation of first impressions [30,108,110]. By contrast, evidence that first impressions lack accuracy and vary between cultural groups accords well with the view that first impressions are culturally acquired [43,100,104]. As described above, however, the focus on White faces and the use of predominantly White participants has artificially exaggerated evidence of accuracy and consistency, thereby 'stacking the deck' in favour of nativist accounts of first impressions.

We note that the prevailing focus on White faces has also obscured some important, and potentially troubling, questions posed by the nativist view: Do humans possess innate representations about White faces and faces of colour? Do people who identify as White, Black and Asian possess similar innate representations? Do racial stereotypes have an innate basis? A constructive discussion about the relationship between racial stereotypes and other types of first impression will help authors engage with these questions.

## 4.2. Social consequences

Next, we consider the broader social consequences of the lack of facial diversity in first impressions research. There is increasing recognition that cognitive science needs to move beyond a focus on White and WEIRD (Western, educated, industrialized, rich and democratic) populations [111]. If the study of first impressions does not incorporate greater diversity in the facial images used, there is a danger that researchers will inadvertently reinforce the idea that White faces are the 'standard'. This is particularly problematic when authors employ only White stimuli, but generalize their findings to all faces without

the appropriate qualifications and caveats. The implication is that White faces are the 'most representative' or 'most typical' kind of human face. The prevailing focus on White face stimuli might also give the impression that understanding first impressions of White faces is a more important or a more interesting research question than understanding first impressions of Black or Asian faces.

More generally, scientists have a particular responsibility when talking about 'race'. The lay belief that racial categories reflect biological differences is dangerous as it can be used to justify status differences between groups [112]. It is important, therefore, that authors in this field acknowledge appropriately (i) that perceived racial groups are in large part social constructions, and (ii) that people do not fall neatly into one discrete racial category or another [113]. In reality the divide between 'Black' and 'White' individuals, for example, owes at least as much to cultural factors as to biology: two people, one who identifies as Black and one who identifies as White, could easily share more genes than two people who identify as Black [114,115]. Many people identify as biracial or multiracial—including a number of well-known celebrities (e.g. Vin Diesel, Tim Howard, Halle Berry, Mariah Carey, Keanu Reeves, Barack Obama) and it is not always possible to tell how someone identifies based solely on their appearance.

During peer review, it has been put to us that: (i) the public do not read scientific papers and certainly do not pay close attention to the stimuli used, and (ii) that there is little or no 'trickle-down' from scientific discourse into the public consciousness. As such, our concerns about the potential social consequences of the lack of facial diversity in the first impressions literature are over-stated. It is important to appreciate, however, that this topic receives a great deal of coverage in the popular media, and articles frequently include images from the studies referenced [116–119]. When reading some of this coverage (e.g. [117]), it would be easy to conclude that understanding the first impressions of White faces is a higher priority for scientists than understanding first impressions formed about faces of colour.

## 4.3. Missed opportunities for social change

Much has been written about the systematic bias that some White individuals encounter because of their facial appearance. Research suggests that White people with untrustworthy appearance are less likely to be hired for jobs, may receive harsher sentencing decisions, and may struggle to borrow money [2,4,12]. Recent work has, therefore, examined if and how first impressions can be modified. Studies in the laboratory suggest that participants readily learn new face-trait mappings, and generalize that learning to new faces of similar appearance [23,120,121]. Similarly, authors are exploring potential interventions to reduce the damaging social consequences of faceism [12].

People who identify as White may well experience real-world problems as a result of faceism. In the vast majority of cases, however, these problems are far less damaging than the systematic racism experienced by particular groups on the basis of perceived ethnicity. In the US, Europe, and elsewhere, people of colour face systematic discrimination in employment, criminal justice settings and in the political sphere [46,50]. Far too often, first impressions made about people of colour have fatal consequences [46,61,85]. It would be a travesty if opportunities to leverage the knowledge derived from the study of first impressions against the dire consequences of racial bias were lost due to ill-conceived and arbitrary choices about stimuli.

# 5. Positive developments

To finish, we want to highlight some positive developments in first impressions research. Some authors have started to ask whether results obtained using White faces replicate when stimulus sets include faces of colour [30,56,58,122–124]. A particularly good example is a recent study described by Jones et al. [56]. The aim of this work was to assess the extent to which a two factor (dominance-trustworthiness) model summarizes the trait evaluations made by people from different cultures around the world. Participants were asked to rate the traits of a diverse stimulus set comprising 30 Black faces, 30 White faces, 30 Asian faces and 30 Latin faces. We see no reason why this level of stimulus diversity should not be routinely used in future first impressions research.

We also wish to acknowledge empirical and theoretical work that has started to address the relationship between racial biases and stereotypes and first impressions. As described above, Todorov et al. [2] argued that faceism has a perceptual basis, whereas the effects of inter-group stereotyping have a non-perceptual basis and are, therefore, qualitatively different. While we disagree with the

authors, we find this kind of explicit discussion to be extremely constructive (see also [54,57]). We also wish to highlight previous fMRI investigations of the neural responses when observers make trait evaluations of individuals of different ethnicities (e.g. [81]). It would be valuable to see more work that directly compares the neural substrate of racial biases and other types of trait inference. As has been noted previously [54], there appears to be some overlap between the neural responses when viewing same-ethnicity faces deemed untrustworthy, and other-ethnicity faces associated with negative stereotypes [21,24,125].

# 6. Conclusion

It is the convention in the study of first impressions from faces to remove cues to ethnicity from the facial stimuli used. Typically, this means that participants are asked to judge the likely traits of White faces only. Frequently, little or no explanation is offered for the lack of facial diversity. Having considered four possible reasons for this convention, however, we find little justification for the lack of facial diversity in this literature. We have argued that the narrow focus on White faces has likely hindered scientific efforts to understand first impressions from faces and may have had unintended and socially deleterious consequences.

We believe attempts to generalize findings obtained with White faces to all faces has caused systematic over-estimation of judgement validity and inter-rater consensus. Consequently, the empirical support for nativist accounts of first impressions and claims of universality is not as strong as it appears. Where authors continue to use ethnically homogeneous stimuli, it is crucial that they adequately reflect the consequences of this narrow subsampling of human faces in the conclusions they draw. In particular, arguments about evolved capacities cannot be based solely on the study of White people who represent a minority of the world's population.

More generally, there has been a lack of explicit discussion about the relationship between group stereotypes and other types of first impression. Where researchers maintain that racist facial stereotypes are qualitatively different from other types of first impressions, and consequently that ethnically homogeneous stimuli are justified, it is critical that they make their rationale explicit. Without a transparent and constructive conversation, differences in the implicit assumptions made by authors may make it increasingly difficult to evaluate rival theoretical perspectives.

The presence of greater diversity in stimulus sets may produce some complex results. There may be evidence of contextual variability and individual differences, and some feature–trait relationships may vary as a function of perceived face ethnicity. However, this is not a problem with the approach *per se*. Rather, this additional complexity reflects the reality of the phenomenon.

Ethics. R.C. and H.O. wrote the manuscript. Both authors gave approval for publication and agree to be held accountable for the work performed therein.

Data accessibility. This article has no additional data.

Competing interests. We declare we have no competing interests.

Funding. This research was supported by the European Research Council under the European Union's Horizon 2020 Programme, grant nos. ERC-STG-715824 and ERC-STG-755719. H.O. was supported by a Philip Leverhulme Prize.

Acknowledgements. We thank Katie Gray, Cade McCall, Steve Tipper, Maria Tsantani and Daniel Yon for valuable comments on an earlier draft.

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
