## [Peer Review File · Royal Society Open Science]

Review History

RSOS-202113.R0 (Original submission)

Review form: Reviewer 1

Is the manuscript scientifically sound in its present form?

Yes

Are the interpretations and conclusions justified by the results?

Yes

Is the language acceptable?

Yes

Do you have any ethical concerns with this paper?

No

Have you any concerns about statistical analyses in this paper?

No

Recommendation?

Accept with minor revision (please list in comments)

Comments to the Author(s)

Both authors are experts on the topic of first impressions from faces, the manuscript is interesting and very well written, and I agree with (most of) its content.

This being said, I would like to invite the authors to discuss a potential point of concern with their proposition of increasing the amount and the number of differences between facial stimuli in studies on first impressions.

The authors focus on the fact that stereotypes can influence first impressions of faces, so to say in a “top-down” way. From this perspective, the sight of a face activates numerous stereotypes (e.g., of gender), even when only white faces are used. Including more racially diverse faces therefore should not, according to the authors, introduce anything that is qualitatively different. On the contrary, they argue that it should enrich the field of first impressions studies.

However, a point that the authors seem not to have addressed is that associations between facial features and specific mental representations may sometimes operate “bottom-up”, due to morphological resemblance. For example, facial expressions of anger show more visual resemblance with the morphological features of average male faces with neutral expressions – typically characterised by prominent eyebrows, thinner lips, and a more squared jaw – than to the morphological features of average female faces (Said et al., 2009; Zebrowitz et al., 2010).

This begs the question, if the visual resemblance between faces at rest and specific emotional facial expressions also varies between ethnic groups (either in absolute terms, or in the context of the observer’s own ethnic background/familiarity). Assuming that such an association between ethnic background and facial expressions exists, would we not want to control for it? Neutral and emotional facial expressions result in different brain responses after just 100ms (Vuilleumier & Pourtois, 2007). If faces from diverse ethnic backgrounds present different degrees of visible resemblance with emotional facial expressions, then striving for greater ethnic stimulus variability might induce unwanted “bottom-up” effects on behavioural and brain responses, which are distinct (independent or correlated, we do not know yet) from the effects due to stereotype activation discussed by Cook and Over.

Early event related potentials (ERPs) have also been shown to be sensitive to aspects of visual stimuli that typically lie outside researchers’ interests, such as the power in different spatial frequency bands (Delplanque et al., 2007). Careful control of low-level stimulus features (luminance, spatial frequencies) is thus crucial for well-executed ERP studies, as well as for other methods, such as continuous flash suppression (CFS; Gray et al., 2013; Yang & Blake, 2012). While methodological constraints should not prevent us from increasing ethnic diversity in face stimulus sets, the challenges associated with it should nevertheless be acknowledged.

In summary, the current manuscript makes a convincing claim for the need to include more ethnically diverse stimuli in studies investigating first impressions from faces. The authors however might want to also address that greater stimulus diversity bears the risk of complicating the task of acquiring meaningful measures of brain activity, such as EEG/ERPs, as well as using faces in some paradigms like CFS.

Minor:

The first 2 sentences in the abstract and introduction are identical. Consider changing slightly.

References:

- Delplanque, S., N'Diaye, K., Scherer, K. R., & Grandjean, D. (2007). Spatial frequencies or emotional effects? A systematic measure of spatial frequencies for IAPS pictures by a discrete wavelet analysis. *Journal of Neuroscience Methods*, 165(1), 144–150.
- Gray, K. L. H., Adams, W. J., Hedger, N., Newton, K. E., & Garner, M. (2013). Faces and awareness: Low-level, not emotional factors determine perceptual dominance. *Emotion (Washington, D.C.)*, 13(3), 537–544. <https://doi.org/10.1037/a0031403>
- Said, C. P., Sebe, N., & Todorov, A. (2009). Structural resemblance to emotional expressions predicts evaluation of emotionally neutral faces. *Emotion*, 9(2), 260–264. <https://doi.org/10.1037/a0014681>
- Vuilleumier, P., & Pourtois, G. (2007). Distributed and interactive brain mechanisms during emotion face perception: Evidence from functional neuroimaging. *Neuropsychologia*, 45(1), 174–194.
- Yang, E., & Blake, R. (2012). Deconstructing continuous flash suppression. *Journal of Vision*, 12(3), 8. <https://doi.org/10.1167/12.3.8>
- Zebrowitz, L. A., Kikuchi, M., & Fellous, J.-M. (2010). Facial resemblance to emotions: Group differences, impression effects, and race stereotypes. *Journal of Personality and Social Psychology*, 98(2), 175–189. <https://doi.org/10.1037/a0017990>

Review form: Reviewer 2

Is the manuscript scientifically sound in its present form?

No

Are the interpretations and conclusions justified by the results?

No

Is the language acceptable?

Yes

Do you have any ethical concerns with this paper?

No

Have you any concerns about statistical analyses in this paper?

No

Recommendation?

Major revision is needed (please make suggestions in comments)

Comments to the Author(s)

In the current review, the authors review the problematic tendency in the literature of the intuitive trait judgments of strangers from facial appearances ("first impressions") to use only White faces -- often all male and young. They review the rationale for this tendency described in the literature, conclude that these rationale does not withstand scrutiny, and posit including non-White faces in the investigation of the issue should be the norm, not an exception.

I read the manuscript with great interest. While I agree with the general point of the paper, I do have minor but serious concerns regarding specific points found in the paper. I'd like to suggest revisions so that the paper makes its point more clear.

1. Overall, I am under an impression that the review paints a rather pigeonholed picture of the field, which arguably is unfair to many researchers that care about the issue of face diversity. I understand that the pigeonholing rhetoric is an effective way to convey the main idea of the paper, and I agree that in some individual empirical papers cited here, the reasons behind using only White faces are unjustifiable, for the very reasons the authors state. However, as the field began to care more about the issue of face diversity in the face sample, recent papers now use more comprehensive, diverse set of faces (as partially described in "Positive Signs" by the authors). It should be noted that these are often the same research teams (i.e., teams led by the same authors) that once used only White faces in their older papers. So, it is clear that researchers, our field as a whole, now care about the diversity in face samples.

2. Speaking of old papers, I'd like to take Oosterhof & Todorov (2008) for example. This is a paper published 13 years ago, and is admittedly one of the first papers that investigated the issue of the structure of social evaluation of others' faces (and its underlying perceptual basis). This in my opinion is already a daunting task, and at that point it makes sense at least to me to keep one's stimuli homogeneous, racially and otherwise. You got to start from somewhere. In the light of this, it is equally important to emphasize how the field reacted to the paper (and to similar papers that followed the paper). Naturally, the 2008 paper facilitated future research to look into the variabilities in the structure of face evaluation across face categories – e.g., races, age groups, sexes:

- races: Sutherland et al. (2018) built separate data-driven evaluation models of East Asian and Caucasian male/female faces recruiting East Asian and Caucasian participants

- age groups: Collova et al. (2019) found for children's faces, unlike the original 2d model (Oosterhof & Todorov, 2008), two summary dimensions of "valence" and "shyness" implying that the face evaluation structure may be driven by social goals in the perceiver's mind (e.g., interactions with the a grown-up vs. child).

- sexes: Oh et al. (2019) extracted separate summary dimensions of male and female faces via separate PCAs/correlations of the face-based trait ratings, and built separate data-driven trait models of male and female faces.

These later efforts highlight that we as a field are not stuck in a regressive mindset of using only one kind of (i.e., White) faces. The "Positive Signs" section definitely echoes the same sentiment and is great for a balanced discussion, but I think how expansive the change in the field is right now, and how seriously people are considering the issue could be better expressed.

3. p.12 "particularly good example is a recent study described by Jones and colleagues ... the authors found that cross-cultural differences have been under-estimated in the extant literature."

I concur that Jones et al. (2021) is an admirable project with much value. I am glad that the study included Asian, Black, and Latino faces in their stimulus set. However, I have to say this particular statement by the authors reflects an incorrect reading of the paper. Jones et al. found highly similar structure across all regions when they applied a PCA to different world regions (the same method used by the Oosterhof and Todorov's original 2008 paper) – arguably except for the Eastern Europe (see Table 3). The previously "underestimated" cross-cultural differences emerged only due to a different *method* applied to the data between the paper and the original 2008 paper. (i.e., PCA vs. factor analysis), not different *world regions*. Moreover, the cross-regional differences may be inflated due to a methodological artifact, a recent paper from the Todorov's lab (Todorov & Oh 2021) argues, and when the artifact is adjusted, there appears to be virtually no cross-regional differences in the data.

I would like to emphasize this: It is absolutely possible to celebrate the fact that the 2021 paper employed racially diverse facesets, while admitting that there was little cross-regional difference found in the Valence-Power face evaluation structure (after all, you never know what you'll find out until you actually test your ideas). On the other hand, weighted indexes of Valence and Power of individual faces (not the Valence-Power structure per se) may be able to reveal some cross-regional differences in people's impressions of racially diverse faces. This was revealed using the same international, Jones et al. data (Todorov & Oh 2021; see Section 1.3 and Figure 6). So there may be indeed some cross-regional variabilities (induced by the racially diverse faces) there.

4. There were several statements in the paper that lack citations that can benefit from including the exact sources.

"in reality, the divide between "black" and "white" faces owes at least as much to cultural factors as to biology: two people, one who identifies as black and one who identifies as white, could easily share more genes than two people who identify as black." : I really want this to be true. However, my knowledge (or acceptance thereof) should be based on data, not my feelings. Thus, I'm keen to see a citation on this.

"Moreover, not everyone shows other-race effects. People are thought to develop expertise for the types of faces to which they are exposed. Where first impressions research conducted in diverse urban centres (e.g. New York, London), it seems unjustified to assume that local participants lack perceptual expertise for faces of colour." : Are these sentences a conjecture or empirical truth? The first sentence ("not everyone shows other-race effects") is almost certainly true given the wide individual variabilities in face processing, but it needs a citation. "it's unjustifiable to assume that residents of New York or London lack perceptual expertise for faces of colour" Expertise in face categories seems to require so much more than casual encounters with other categories. I'm happy to be corrected, and I'd like to see a citation.

5. As you may be already aware, it's much preferable to write "Black" and "White" with the capital B and W's (vs. b's and w's) for cultural/historical reasons. Use of Black vs. black (and White vs white) seems inconsistent in the paper (e.g., on p.6 there are many "black faces"s). I suggest "Black" and "White" throughout.

6. In Figures 1, there's a letter "d" inside the panel a.

Review form: Reviewer 3

Is the manuscript scientifically sound in its present form?

Yes

Are the interpretations and conclusions justified by the results?

Yes

Is the language acceptable?

Yes

Do you have any ethical concerns with this paper?

No

Have you any concerns about statistical analyses in this paper?

No

Recommendation?

Major revision is needed (please make suggestions in comments)

Comments to the Author(s)

This paper gives some consideration to the lack of racial diversity in the first impressions literature. The authors assert that the field has traditionally focused on white raters evaluating white faces, often without justification, and argue for the inclusion of more racially diverse stimuli. They describe some common justifications for using ethnically homogeneous stimuli—for instance, the concern that inferences from ‘ethnicity cues’ and ‘morphological face cues’ are qualitatively different, or that diverse faces distort findings by introducing noise (e.g., stemming from a lack of perceptual expertise viewing other-race faces). The authors argue that these justifications do not hold up to scrutiny, for example observing that many studies use computer-generated stimuli with which participants have little perceptual expertise—yet are still considered useful.

Overall, the paper is well-written, and I agree with the authors’ thesis that the lack of facial diversity is a problem. Although I found some of their arguments more compelling than others, I agree with the message that emphasizing White faces means missed opportunities for understanding how impressions are formed across group boundaries.

Below I have some comments and questions that I hope to see the authors address.

Major Considerations:

1) The main issue addressed in this review paper is important. Although I agree with the authors’ thesis, I very much struggled with the central premise that the focus on white faces has “served to reinforce socially regressive ideas about ‘race’”. The authors repeatedly make this argument throughout the paper, for example on page 9, where the authors speculate that “the focus on white faces may have also had damaging social consequences. It may have inadvertently reinforced the regressive idea that there are discrete racial categories each with a particular genetic profile.” This argument seems to hinge on the assumption that the public either reads scientific papers about impression formation or pays close attention to the stimuli used in these studies. Or that scientific discourse has otherwise trickled into public consciousness. I have some doubts about whether that’s the case, and how much the use of stimuli in face perception research might really be contributing to the broader perception of race as categories. If the authors wish to maintain this argument, I’d be interested in empirical evidence supporting their position. But I think the authors can make the same arguments without including this issue at all, and it would not detract from the value of their paper.

2) At times, I felt the authors misrepresented some of the literature which hampered my overall enthusiasm of this work. For example, the last paragraph beginning on page 8 implies that viewing first impressions as a “genetic adaptation” might lead to an “implausible, and deeply offensive” perspective that “first impressions of Black men as untrustworthy and aggressive are accurate”. I wonder if the authors have conflated adaptive and accurate here. Ecological theories of face perception assert that the ability to quickly extract social information from faces is adaptive (Zebrowitz, 2006), not necessarily accurate. That is, from a “risk management” perspective, it may be more adaptive to (incorrectly) judge an outgroup member as untrustworthy than to (incorrectly) judge an outgroup member as trustworthy (Zebrowitz, Fellous, Mignault & Adreolletti, 2003; Zebrowitz, Kikuchi, & Fellous, 2010). As such, facial impressions may have a functional basis even if they are biased or inaccurate.

It also seems like an odd segue from the previous paragraph, where the authors talk about the accuracy of trustworthiness judgments, but then jump to negative racial stereotypes about trustworthiness. Given that trustworthy might mean different things in different contexts (Wilson & Rule, 2017), I believe this paragraph could be improved by simply stating that humans aren't very accurate at judging trustworthiness, and that there's an abundance of literature to demonstrate that (see Todorov, Olivola, Dotsch, & Mende-Siedlecki, 2015).

3) I was not convinced by the argument that just because White faces also elicit group stereotypes like "jock" and "geek" (end of page 6), that group stereotypes "within" the same "race" are qualitatively the same as group stereotypes "between races". While categorical thinking may be inevitable, some researchers may want to randomize out top-down categorical effects (e.g., stereotyping) as much as they can. For example, while it's possible that a group of White faces will elicit stereotypes like "plant lover", "geeks", "jocks", "hipsters", etc., it's far more likely that the stimulus set will randomly sample from a universe of (White) people stereotypes. However, if you use a stimulus set comprising White, Black, and Korean faces, you are not randomly sampling from a universe of racial groups but emphasizing the specific racial categories in the stimulus set. It seems likely that emphasizing these racial categories would induce stronger top-down effects of racial stereotypes. Researchers who want to minimize top-down categorical effects as much as possible would have reason to avoid using mixed-race stimuli in this way, or use alternative data collection designs (i.e., blocking by race) to avoid this issue.

4) Somewhat related to the above comment, I think the strongest argument for including racially diverse stimuli is that it's important to test whether findings generalize to people from other racial groups. Any model that's developed on White faces, if intended to be universally applicable, should fit well for other groups. The authors do mention that an influential model of face perception does not readily generalize to other cultures (Jones et al., 2021), and I expected this to be framed as an important demonstration of the value of using racially diverse stimuli. Although this is just my opinion, I consider this to be the most important point, and thought it could be emphasized more.

5) I appreciate that the authors have cited broadly to demonstrate the overreliance on white faces in the first impressions literature. However, a growing body of work does use more racially diverse stimuli, some of which the authors cite (Jones et al., 2021) and some they do not. Some of these papers speak directly to the claims made by the authors, and rather than speculate, they could just cite the existing evidence.

For example, the authors can back up their speculation on page 10 that "Greater diversity in stimulus sets is therefore likely to reveal more heterogeneous first impressions" by citing Xie, Flake, and Hehman (2019), which uses a large, racially diverse stimulus set to examine the variability in facial impressions across different race and gender groups. Regardless of target race, "perceiver idiosyncrasies" contribute more variance to face impressions (~25%) than do "shared consensus" (~15%), providing empirical support for the claim that any consensus in facial judgments might be overstated when participants are evaluating racially diverse faces.

A notable omission is a recent paper by Stolier, Hehman, Keller, Walker, and Freeman (2018), which finds that perceivers' own lay theories of personality (trait space) map onto the structure of their facial impressions (face space) even when participants are only evaluating white male stimuli. This speaks directly to the authors' argument that trait inferences from facial cues may be shaped by participants' experiences, and is also relevant to the argument that top-down processing occurs even in racially homogeneous stimuli sets.

Minor Considerations:

6) While this is a paper about face perception, the first sentence is too strong and somewhat incorrect, and I interpreted it as a typo. The authors state, "When we first encounter a stranger, we spontaneously attribute to them a wide variety of character traits based solely on their facial appearance." As people use other cues to form an impression (e.g., body, clothes, hairstyle), I suggest removing "solely" from this sentence.

7) When discussing the 'other-race effect', the authors talk about it entirely from the perceptual expertise account but do not mention motivational accounts of other-race effects. Can the authors consider the relevance of this motivational component and how it might impact their conclusions? From this perspective, familiarity with the outgroups likely wouldn't matter for impacting trait impressions from faces (see Young, Hugenberg, Bernstein, & Sacco, 2012).

See:

Young, S. G., Hugenberg, K., Bernstein, M. J., & Sacco, D. F. (2012). Perception and motivation in face recognition: A critical review of theories of the cross-race effect. *Personality and Social Psychology Review*, 16(2), 116-142.

8) Somewhat related to an earlier point, presenting racially heterogeneous stimuli to participants may change the way that they anchor their impressions. Is there a difference between using a racially diverse stimuli set presented in a racially homogeneous way (e.g., all Black, all White, between-participants) versus presenting a racially heterogeneous presentation (50% Black, 50% White)? What are some recommendations the authors can make for including diverse stimuli in studies (e.g., separate batches, mixed in equal amounts, mixed in some other way)? I wasn't sure what evidence was out there that might speak to this.

9) I appreciated that the authors pointed readers to several databases with more racially diverse stimuli, and would like to suggest a few more that the authors might be unaware of.

Chen, J. M., Norman, J. B., & Nam, Y. (2020). Broadening the Stimulus Set: Introducing the American Multiracial Faces Database. *Behavior Research Methods*, 1-19.

DeBruine, L. M., & Jones, B. C. (2017). Face Research Lab London Set. figshare. <https://doi.org/10.6084/m9.figshare.5047666>

Strohmingner, N., Gray, K., Chituc, V., Heffner, J., Schein, C., & Heagins, T. B. (2016). The MR2: A multi-racial, mega-resolution database of facial stimuli. *Behavior research methods*, 48(3), 1197-1204.

Decision letter (RSOS-202113.R0)

Dear Dr Cook

The Editors assigned to your paper RSOS-202113 "Why is the literature on first impressions so focused on white faces?" have made a decision based on their reading of the paper and any comments received from reviewers.

Regrettably, in view of the reports received, the manuscript has been rejected in its current form. However, a new manuscript may be submitted which takes into consideration these comments.

We invite you to respond to the comments supplied below and prepare a resubmission of your manuscript. Below the referees' and Editors' comments (where applicable) we provide additional requirements. We provide guidance below to help you prepare your revision.

Please note that resubmitting your manuscript does not guarantee eventual acceptance, and we do not generally allow multiple rounds of revision and resubmission, so we urge you to make every effort to fully address all of the comments at this stage. If deemed necessary by the Editors, your manuscript will be sent back to one or more of the original reviewers for assessment. If the original reviewers are not available, we may invite new reviewers.

Please resubmit your revised manuscript and required files (see below) no later than 23-Aug-2021. Note: the ScholarOne system will 'lock' if resubmission is attempted on or after this deadline. If you do not think you will be able to meet this deadline, please contact the editorial office immediately.

Please note article processing charges apply to papers accepted for publication in Royal Society Open Science (<https://royalsocietypublishing.org/rsos/charges>). Charges will also apply to papers transferred to the journal from other Royal Society Publishing journals, as well as papers submitted as part of our collaboration with the Royal Society of Chemistry (<https://royalsocietypublishing.org/rsos/chemistry>). Fee waivers are available but must be requested when you submit your manuscript (<https://royalsocietypublishing.org/rsos/waivers>).

Thank you for submitting your manuscript to Royal Society Open Science and we look forward to receiving your resubmission. If you have any questions at all, please do not hesitate to get in touch.

on behalf of Dr Giorgia Silani (Associate Editor) and Essi Viding (Subject Editor)
openscience@royalsociety.org

Associate Editor Comments to Author (Dr Giorgia Silani):

Associate Editor: 1

Comments to the Author:

The paper has now been reviewed by three reviewers. While they all found your work to be potentially important for RSOS, they have highlighted some weakness in the conceptual arguments and misinterpretation of the cited literature. They have provided constructive suggestions that would need to be addressed before the manuscript would be considered for publication. We believe that the reject and resubmit option will give you more time to address such comments.

Reviewer comments to Author:

Reviewer: 1

Comments to the Author(s)

Both authors are experts on the topic of first impressions from faces, the manuscript is interesting and very well written, and I agree with (most of) its content.

This being said, I would like to invite the authors to discuss a potential point of concern with their proposition of increasing the amount and the number of differences between facial stimuli in studies on first impressions.

The authors focus on the fact that stereotypes can influence first impressions of faces, so to say in a “top-down” way. From this perspective, the sight of a face activates numerous stereotypes (e.g., of gender), even when only white faces are used. Including more racially diverse faces therefore should not, according to the authors, introduce anything that is qualitatively different. On the contrary, they argue that it should enrich the field of first impressions studies.

However, a point that the authors seem not to have addressed is that associations between facial features and specific mental representations may sometimes operate “bottom-up”, due to morphological resemblance. For example, facial expressions of anger show more visual resemblance with the morphological features of average male faces with neutral expressions – typically characterised by prominent eyebrows, thinner lips, and a more squared jaw – than to the morphological features of average female faces (Said et al., 2009; Zebrowitz et al., 2010).

This begs the question, if the visual resemblance between faces at rest and specific emotional facial expressions also varies between ethnic groups (either in absolute terms, or in the context of the observer’s own ethnic background/familiarity). Assuming that such an association between ethnic background and facial expressions exists, would we not want to control for it? Neutral and emotional facial expressions result in different brain responses after just 100ms (Vuilleumier & Pourtois, 2007). If faces from diverse ethnic backgrounds present different degrees of visible resemblance with emotional facial expressions, then striving for greater ethnic stimulus variability might induce unwanted “bottom-up” effects on behavioural and brain responses, which are distinct (independent or correlated, we do not know yet) from the effects due to stereotype activation discussed by Cook and Over.

Early event related potentials (ERPs) have also been shown to be sensitive to aspects of visual stimuli that typically lie outside researchers’ interests, such as the power in different spatial frequency bands (Delplanque et al., 2007). Careful control of low-level stimulus features (luminance, spatial frequencies) is thus crucial for well-executed ERP studies, as well as for other methods, such as continuous flash suppression (CFS; Gray et al., 2013; Yang & Blake, 2012). While methodological constraints should not prevent us from increasing ethnic diversity in face stimulus sets, the challenges associated with it should nevertheless be acknowledged.

In summary, the current manuscript makes a convincing claim for the need to include more ethnically diverse stimuli in studies investigating first impressions from faces. The authors however might want to also address that greater stimulus diversity bears the risk of complicating the task of acquiring meaningful measures of brain activity, such as EEG/ERPs, as well as using faces in some paradigms like CFS.

Minor:

The first 2 sentences in the abstract and introduction are identical. Consider changing slightly.

References:

- Delplanque, S., N'Diaye, K., Scherer, K. R., & Grandjean, D. (2007). Spatial frequencies or emotional effects? A systematic measure of spatial frequencies for IAPS pictures by a discrete wavelet analysis. *Journal of Neuroscience Methods*, 165(1), 144–150.
- Gray, K. L. H., Adams, W. J., Hedger, N., Newton, K. E., & Garner, M. (2013). Faces and awareness: Low-level, not emotional factors determine perceptual dominance. *Emotion (Washington, D.C.)*, 13(3), 537–544. <https://doi.org/10.1037/a0031403>
- Said, C. P., Sebe, N., & Todorov, A. (2009). Structural resemblance to emotional expressions predicts evaluation of emotionally neutral faces. *Emotion*, 9(2), 260–264. <https://doi.org/10.1037/a0014681>
- Vuilleumier, P., & Pourtois, G. (2007). Distributed and interactive brain mechanisms during emotion face perception: Evidence from functional neuroimaging. *Neuropsychologia*, 45(1), 174–194.
- Yang, E., & Blake, R. (2012). Deconstructing continuous flash suppression. *Journal of Vision*, 12(3), 8. <https://doi.org/10.1167/12.3.8>
- Zebrowitz, L. A., Kikuchi, M., & Fellous, J.-M. (2010). Facial resemblance to emotions: Group differences, impression effects, and race stereotypes. *Journal of Personality and Social Psychology*, 98(2), 175–189. <https://doi.org/10.1037/a0017990>

Reviewer: 2

Comments to the Author(s)

In the current review, the authors review the problematic tendency in the literature of the intuitive trait judgments of strangers from facial appearances ("first impressions") to use only White faces -- often all male and young. They review the rationale for this tendency described in the literature, conclude that this rationale does not withstand scrutiny, and posit including non-White faces in the investigation of the issue should be the norm, not an exception.

I read the manuscript with great interest. While I agree with the general point of the paper, I do have minor but serious concerns regarding specific points found in the paper. I'd like to suggest revisions so that the paper makes its point more clear.

1. Overall, I am under an impression that the review paints a rather pigeonholed picture of the field, which arguably is unfair to many researchers that care about the issue of face diversity. I understand that the pigeonholing rhetoric is an effective way to convey the main idea of the paper, and I agree that in some individual empirical papers cited here, the reasons behind using only White faces are unjustifiable, for the very reasons the authors state. However, as the field began to care more about the issue of face diversity in the face sample, recent papers now use more comprehensive, diverse sets of faces (as partially described in "Positive Signs" by the authors). It should be noted that these are often the same research teams (i.e., teams led by the same authors) that once used only White faces in their older papers. So, it is clear that researchers, our field as a whole, now care about the diversity in face samples.

2. Speaking of old papers, I'd like to take Oosterhof & Todorov (2008) for example. This is a paper published 13 years ago, and is admittedly one of the first papers that investigated the issue of the structure of social evaluation of others' faces (and its underlying perceptual basis). This in my opinion is already a daunting task, and at that point it makes sense at least to me to keep one's stimuli homogeneous, racially and otherwise. You got to start from somewhere. In the light of this, it is equally important to emphasize how the field reacted to the paper (and to similar papers that followed the paper). Naturally, the 2008 paper facilitated future research to look into the variabilities in the structure of face evaluation across face categories – e.g., races, age groups, sexes:

- races: Sutherland et al. (2018) built separate data-driven evaluation models of East Asian and Caucasian male/female faces recruiting East Asian and Caucasian participants

- age groups: Collova et al. (2019) found for children's faces, unlike the original 2d model (Oosterhof & Todorov, 2008), two summary dimensions of "valence" and "shyness" implying that the face evaluation structure may be driven by social goals in the perceiver's mind (e.g., interactions with the a grown-up vs. child).

- sexes: Oh et al. (2019) extracted separate summary dimensions of male and female faces via separate PCAs/correlations of the face-based trait ratings, and built separate data-driven trait models of male and female faces.

These later efforts highlight that we as a field are not stuck in a regressive mindset of using only one kind of (i.e., White) faces. The "Positive Signs" section definitely echoes the same sentiment and is great for a balanced discussion, but I think how expansive the change in the field is right now, and how seriously people are considering the issue could be better expressed.

3. p.12 "particularly good example is a recent study described by Jones and colleagues ... the authors found that cross-cultural differences have been under-estimated in the extant literature."

I concur that Jones et al. (2021) is an admirable project with much value. I am glad that the study included Asian, Black, and Latino faces in their stimulus set. However, I have to say this particular statement by the authors reflects an incorrect reading of the paper. Jones et al. found highly similar structure across all regions when they applied a PCA to different world regions (the same method used by the Oosterhof and Todorov's original 2008 paper) – arguably except for the the Eastern Europe (see Table 3). The previously "underestimated" cross-cultural differences emerged only due to a different *method* applied to the data between the paper and the original 2008 paper. (i.e., PCA vs. factor analysis), not different *world regions*. Moreover, the cross-regional differences may be inflated due to a methodological artifact, a recent paper from the Todorov's lab (Todorov & Oh 2021) argues, and when the artifact is adjusted, there appears to be virtually no cross-regional differences in the data.

I would like to emphasize this: It is absolutely possible to celebrate the fact that the 2021 paper employed racially diverse facesets, while admitting that there was little cross-regional difference found in the Valence-Power face evaluation structure (after all, you never know what you'll find out until you actually test your ideas). On the other hand, weighted indexes of Valence and Power of individual faces (not the Valence-Power structure per se) may be able to reveal some cross-regional differences in people's impressions of racially diverse faces. This was revealed using the same international, Jones et al. data (Todorov & Oh 2021; see Section 1.3 and Figure 6). So there may be indeed some cross-regional variabilities (induced by the racially diverse faces) there.

4. There were several statements in the paper that lack citations that can benefit from including the exact sources.

"in realty, the divide between "black" and "white" faces owes at least as much to cultural factors as to biology: two people, one who identifies as black and one who identifies as white, could easily share more genes than two people who identify as black." : I really want this to be true. However, my knowledge (or acceptance thereof) should be based on data, not my feelings. Thus, I'm keen to see a citation on this.

"Moreover, not everyone shows other-race effects. People are thought to develop expertise for the types of faces to which they are exposed. Where first impressions research conducted in diverse

urban centres (e.g. New York, London), it seems unjustified to assume that local participants lack perceptual expertise for faces of colour." : Are these sentences a conjecture or empirical truth? The first sentence ("not everyone shows other-race effects") is almost certainly true given the wide individual variabilities in face processing, but it needs a citation. "it's unjustifiable to assume that residents of New York or London lack perceptual expertise for faces of colour" Expertise in face categories seems to require so much more than casual encounters with other categories. I'm happy to be corrected, and I'd like to see a citation.

5. As you may be already aware, it's much preferable to write "Black" and "White" with the capital B and W's (vs. b's and w's) for cultural/historical reasons. Use of Black vs. black (and White vs white) seems inconsistent in the paper (e.g., on p.6 there are many "black faces"). I suggest "Black" and "White" throughout.

6. In Figures 1, there's a letter "d" inside the panel a.

Reviewer: 3

Comments to the Author(s)

This paper gives some consideration to the lack of racial diversity in the first impressions literature. The authors assert that the field has traditionally focused on white raters evaluating white faces, often without justification, and argue for the inclusion of more racially diverse stimuli. They describe some common justifications for using ethnically homogeneous stimuli – for instance, the concern that inferences from ‘ethnicity cues’ and ‘morphological face cues’ are qualitatively different, or that diverse faces distort findings by introducing noise (e.g., stemming from a lack of perceptual expertise viewing other-race faces). The authors argue that these justifications do not hold up to scrutiny, for example observing that many studies use computer-generated stimuli with which participants have little perceptual expertise – yet are still considered useful.

Overall, the paper is well-written, and I agree with the authors’ thesis that the lack of facial diversity is a problem. Although I found some of their arguments more compelling than others, I agree with the message that emphasizing White faces means missed opportunities for understanding how impressions are formed across group boundaries.

Below I have some comments and questions that I hope to see the authors address.

Major Considerations:

1) The main issue addressed in this review paper is important. Although I agree with the authors’ thesis, I very much struggled with the central premise that the focus on white faces has “served to reinforce socially regressive ideas about ‘race’”. The authors repeatedly make this argument throughout the paper, for example on page 9, where the authors speculate that “the focus on white faces may have also had damaging social consequences. It may have inadvertently reinforced the regressive idea that there are discrete racial categories each with a particular genetic profile.” This argument seems to hinge on the assumption that the public either reads scientific papers about impression formation or pays close attention to the stimuli used in these studies. Or that scientific discourse has otherwise trickled into public consciousness. I have some doubts about whether that’s the case, and how much the use of stimuli in face perception research might really be contributing to the broader perception of race as categories. If the authors wish to maintain this argument, I’d be interested in empirical evidence supporting their position. But I think the authors can make the same arguments without including this issue at all, and it would not detract from the value of their paper.

2) At times, I felt the authors misrepresented some of the literature which hampered my overall enthusiasm of this work. For example, the last paragraph beginning on page 8 implies that viewing first impressions as a “genetic adaptation” might lead to an “implausible, and deeply offensive” perspective that “first impressions of Black men as untrustworthy and aggressive are accurate”. I wonder if the authors have conflated adaptive and accurate here. Ecological theories of face perception assert that the ability to quickly extract social information from faces is adaptive (Zebrowitz, 2006), not necessarily accurate. That is, from a “risk management” perspective, it may be more adaptive to (incorrectly) judge an outgroup member as untrustworthy than to (incorrectly) judge an outgroup member as trustworthy (Zebrowitz, Fellous, Mignault & Adreoletti, 2003; Zebrowitz, Kikuchi, & Fellous, 2010). As such, facial impressions may have a functional basis even if they are biased or inaccurate.

It also seems like an odd segue from the previous paragraph, where the authors talk about the accuracy of trustworthiness judgments, but then jump to negative racial stereotypes about trustworthiness. Given that trustworthy might mean different things in different contexts (Wilson & Rule, 2017), I believe this paragraph could be improved by simply stating that humans aren’t very accurate at judging trustworthiness, and that there’s an abundance of literature to demonstrate that (see Todorov, Olivola, Dotsch, & Mende-Siedlecki, 2015).

3) I was not convinced by the argument that just because White faces also elicit group stereotypes like “jock” and “geek” (end of page 6), that group stereotypes “within” the same “race” are qualitatively the same as group stereotypes “between races”. While categorical thinking may be inevitable, some researchers may want to randomize out top-down categorical effects (e.g., stereotyping) as much as they can. For example, while it’s possible that a group of White faces will elicit stereotypes like “plant lover”, “geeks”, “jocks”, “hipsters”, etc., it’s far more likely that the stimulus set will randomly sample from a universe of (White) people stereotypes. However, if you use a stimulus set comprising White, Black, and Korean faces, you are not randomly sampling from a universe of racial groups but emphasizing the specific racial categories in the stimulus set. It seems likely that emphasizing these racial categories would induce stronger top-down effects of racial stereotypes. Researchers who want to minimize top-down categorical effects as much as possible would have reason to avoid using mixed-race stimuli in this way, or use alternative data collection designs (i.e., blocking by race) to avoid this issue.

4) Somewhat related to the above comment, I think the strongest argument for including racially diverse stimuli is that it’s important to test whether findings generalize to people from other racial groups. Any model that’s developed on White faces, if intended to be universally applicable, should fit well for other groups. The authors do mention that an influential model of face perception does not readily generalize to other cultures (Jones et al., 2021), and I expected this to be framed as an important demonstration of the value of using racially diverse stimuli. Although this is just my opinion, I consider this to be the most important point, and thought it could be emphasized more.

5) I appreciate that the authors have cited broadly to demonstrate the overreliance on white faces in the first impressions literature. However, a growing body of work does use more racially diverse stimuli, some of which the authors cite (Jones et al., 2021) and some they do not. Some of these papers speak directly to the claims made by the authors, and rather than speculate, they could just cite the existing evidence.

For example, the authors can back up their speculation on page 10 that “Greater diversity in stimulus sets is therefore likely to reveal more heterogenous first impressions” by citing Xie, Flake, and Hehman (2019), which uses a large, racially diverse stimulus set to examine the variability in facial impressions across different race and gender groups. Regardless of target race, “perceiver idiosyncrasies” contribute more variance to face impressions (~25%) than do

“shared consensus” (~15%), providing empirical support for the claim that any consensus in facial judgments might be overstated when participants are evaluating racially diverse faces.

A notable omission is a recent paper by Stolier, Hehman, Keller, Walker, and Freeman (2018), which finds that perceivers’ own lay theories of personality (trait space) map onto the structure of their facial impressions (face space) even when participants are only evaluating white male stimuli. This speaks directly to the authors’ argument that trait inferences from facial cues may be shaped by participants’ experiences, and is also relevant to the argument that top-down processing occurs even in racially homogeneous stimuli sets.

Minor Considerations:

6) While this is a paper about face perception, the first sentence is too strong and somewhat incorrect, and I interpreted it as a typo. The authors state, “When we first encounter a stranger, we spontaneously attribute to them a wide variety of character traits based solely on their facial appearance.” As people use other cues to form an impression (e.g., body, clothes, hairstyle), I suggest removing “solely” from this sentence.

7) When discussing the ‘other-race effect’, the authors talk about it entirely from the perceptual expertise account but do not mention motivational accounts of other-race effects. Can the authors consider the relevance of this motivational component and how it might impact their conclusions? From this perspective, familiarity with the outgroups likely wouldn’t matter for impacting trait impressions from faces (see Young, Hugenberg, Bernstein, & Sacco, 2012).

See:

Young, S. G., Hugenberg, K., Bernstein, M. J., & Sacco, D. F. (2012). Perception and motivation in face recognition: A critical review of theories of the cross-race effect. *Personality and Social Psychology Review*, 16(2), 116-142.

8) Somewhat related to an earlier point, presenting racially heterogeneous stimuli to participants may change the way that they anchor their impressions. Is there a difference between using a racially diverse stimuli set presented in a racially homogeneous way (e.g., all Black, all White, between-participants) versus presenting a racially heterogeneous presentation (50% Black, 50% White)? What are some recommendations the authors can make for including diverse stimuli in studies (e.g., separate batches, mixed in equal amounts, mixed in some other way)? I wasn’t sure what evidence was out there that might speak to this.

9) I appreciated that the authors pointed readers to several databases with more racially diverse stimuli, and would like to suggest a few more that the authors might be unaware of.

Chen, J. M., Norman, J. B., & Nam, Y. (2020). Broadening the Stimulus Set: Introducing the American Multiracial Faces Database. *Behavior Research Methods*, 1-19.

DeBruine, L. M., & Jones, B. C. (2017). Face Research Lab London Set. figshare. <https://doi.org/10.6084/m9.figshare.5047666>

Strohminger, N., Gray, K., Chituc, V., Heffner, J., Schein, C., & Heagins, T. B. (2016). The MR2: A multi-racial, mega-resolution database of facial stimuli. *Behavior research methods*, 48(3), 1197-1204.

===PREPARING YOUR MANUSCRIPT===

===PREPARING YOUR REVISION IN SCHOLARONE===

- An editable file of each table (.doc, .docx, .xls, .xlsx, or .csv).
- An editable file of all figure and table captions.

- Any electronic supplementary material (ESM).
- If you are requesting a discretionary waiver for the article processing charge, the waiver form must be included at this step.
- If you are providing image files for potential cover images, please upload these at this step, and inform the editorial office you have done so. You must hold the copyright to any image provided.
- A copy of your point-by-point response to referees and Editors. This will expedite the preparation of your proof.

- Ensure that your data access statement meets the requirements at <https://royalsociety.org/journals/authors/author-guidelines/#data>. You should ensure that you cite the dataset in your reference list. If you have deposited data etc in the Dryad repository, please include both the 'For publication' link and 'For review' link at this stage.
- If you are requesting an article processing charge waiver, you must select the relevant waiver option (if requesting a discretionary waiver, the form should have been uploaded at Step 3 'File upload' above).
- If you have uploaded ESM files, please ensure you follow the guidance at <https://royalsociety.org/journals/authors/author-guidelines/#supplementary-material> to include a suitable title and informative caption. An example of appropriate titling and captioning may be found at https://figshare.com/articles/Table_S2_from_Is_there_a_trade-off_between_peak_performance_and_performance_breadth_across_temperatures_for_aerobic_scope_in_teleost_fishes_/3843624.

Author's Response to Decision Letter for (RSOS-202113.R0)

See Appendix A.

RSOS-211146.R0

Review form: Reviewer 1

Is the manuscript scientifically sound in its present form?

Yes

Are the interpretations and conclusions justified by the results?

Yes

Is the language acceptable?

Yes

Do you have any ethical concerns with this paper?

No

Have you any concerns about statistical analyses in this paper?

No

Recommendation?

Accept with minor revision (please list in comments)

Comments to the Author(s)

The authors' reply and revision of the manuscript are satisfactory. I only have two further comments.

emotional expressions

In their response to my previous review, the authors write that they "agree that trait evaluations based on the presence of perceived emotion likely differ from those attributable to group or character stereotypes. We now acknowledge this in the revision (see the final paragraph in the section entitled "Does the use of diverse faces introduce a confound?").

The only reference to emotion I could find in that section is in the very last sentence: "If these effects were removed from the literature, together with those effects driven by the (mis)perception of facial emotion [57, 58], we wonder: what would be left?"

I seriously doubt that the average physical resemblance of certain facial features (like those characteristic of male and female faces) with emotional facial expressions can ever be "removed" in the stimuli used so far. Did the authors mean to say that previous research reporting this association only used white faces, and that the association might disappear if a more ethnically diverse set of faces was analysed? If so, maybe they could make this thought more explicit?

implications of showing White faces only

I agree with the other reviewers that it is often advisable to reduce the dimensionality of the stimulus set, at least during initial stages of investigation, to conduct rigorous research. Of course, doing so comes to the expense of ecological validity, and one can ask how much of the results emerging from this way of doing science matter in "real" life. But the same can be said about most other aspects of experimental research, especially the kind being conducted by showing pre-recorded stimuli on computer screens in psychological laboratories. Calls to study social cognition using a more interactive 'second-person' approach have been around for years (Redcay and Schilbach, *Nature Reviews Neuroscience*, 2019), and corresponding efforts to fundamentally change how research in social psychology and neuroscience is conducted are underway.

Although commendable, the authors' invitation to increase the ethnic diversity of face stimuli used in research seems far from sufficient to overcome the fundamental limitation (which is also its main strength) inherent to the scientific method, i.e. its need to isolate specific aspects of a complex phenomenon in order to tightly control them and to rule out as much as possible alternative explanations.

I also don't think that the risk of "overshadowing" the focus of a study by introducing ethnic diversity in the stimulus set can be easily dismissed by using a blocked or between-subjects design - or at least this is an empirical question, i.e. one should first run such a study and then verify that there is no effect of block or group. It is true that "interleaving male and female faces may also render sexually dimorphic cues salient", and that this might influence participants' responses. Arguably, when seeing faces out of context, 'race' is a much more charged concept

than 'gender', and therefore greater "overshadowing" effects can be expected to come from race than gender.

Based on these reflections, I would recommend that the authors town down some of their statements, especially in the abstract. E.g. the sentence "the focus on White faces has undermined scientific efforts to understand first impressions from faces and [...] has served to reinforce socially regressive ideas about 'race'". I think it is somewhat of an exaggeration to say that the focus on White faces has undermined the understanding of first impressions, although I agree that most results obtained so far might not apply to faces from other ethnicities. I also remain sceptical (like Reviewer 3) about the claim that the focus on White faces has reinforced ideas about race, but I appreciate the authors' arguments and accept their view on this.

Review form: Reviewer 2

Is the manuscript scientifically sound in its present form?

Yes

Are the interpretations and conclusions justified by the results?

Yes

Is the language acceptable?

Yes

Do you have any ethical concerns with this paper?

No

Have you any concerns about statistical analyses in this paper?

No

Recommendation?

Accept with minor revision (please list in comments)

Comments to the Author(s)

The authors sufficiently addressed my points, including citing proper previous papers and careful rephrasing of their points. I believe these changes, combined with changes suggested by other reviewers, strengthened the submitted work. I have no remaining issue other than a few minor points:

1. "We also note that certain types of character (e.g., heroes and villains, jocks and geeks) are also prone to stereotypical depiction in film, TV, comics, and storybooks. The activation of these character stereotypes may afford a range of attributions about courage, trustworthiness, and academic and sporting ability."

Perhaps the empirical studies of cultural learning of impression is relevant here as a reference, in which stereotypical depiction of a character affects people's impressions of a group of individuals (e.g., Black individuals; e.g., Weisbuch et al. 2009) including the authors' own work.

Weisbuch, M., Pauker, K., & Ambady, N. (2009). The subtle transmission of race bias via televised nonverbal behavior. *Science*, 326, 1711-1714

2. "More generally, there has been lack of explicit discussion about the relationship a between group stereotypes and other types of first impression." (Conclusion)

Perhaps not the main point here, but to avoid confusion it may be useful to mention that there has been work investigating the relationship between group stereotypes and first impressions. One example is work investigating the perceptual origin of gender/racial face-ism (group stereotypes at the level of faces). For example, different racial groups are often associated with different perceptions [44,45], which are closely emotional perceptions. Some of these associations were found even at the objective visual facial properties (e.g., Zebrowitz et al. (2010) found that White faces resembled angry faces than did East Asian and Black faces; female faces were similar to a surprised face than were male faces. For the category vs. emotion resemblance analysis of the faces, they used connectionist models, not human judges) These findings suggest that perhaps at least partially people are intrinsically biased to form face-based trait impressions of different social categories in a somewhat fixed, different ways. Of course, we should be extra careful when interpreting this line of work, but it is worthy of noting that there are visual properties that lead to category-based face-ism, even without any conceptual stereotypes. I noticed other reviewers raised a similar point.

Also related to said "explicit discussion" is work on the relationship between racial stereotypes and first face impressions. For example, Xie et al. (accepted) found that individual perceivers' different ways of forming impressions of racially diverse faces are related to their self-reported race-related stereotypes, as assessed by second-order similarity analysis.

Both papers are a great example of scientific endeavor examining the relationship a between group stereotypes and other types of first impression (although of course they do not justify any racist narratives or atrocities).

Xie, S. Y., Flake, J. K., Stolier, R. M., Freeman, J. B., & Hehman, E. (conditionally accepted). Facial impressions are predicted by the structure of group stereotypes. *Psychological Science*.
Zebrowitz, L. A., Kikuchi, M., & Fellous, J. M. (2010). Facial resemblance to emotions: group differences, impression effects, and race stereotypes. *Journal of Personality and Social Psychology*, 98(2), 175–189.

3. "Every stimulus face used in the study described by Oh et al. (2019) was White." (Authors' response to R2 #2)

Oh et al. (2019) used racially diverse stimuli to validate their models (Study 3B), which is explicitly stated: "50 photos of eight self-identified East Asian (four females), 8 West Asian (three females), 12 Black (five females), and 22 White actors (13 females) between the ages of 19–37 were used (Supplemental Figure S8 in the online supplementary material). Non-White faces comprised 56% of all faces."

Review form: Reviewer 3

Is the manuscript scientifically sound in its present form?

Yes

Are the interpretations and conclusions justified by the results?

Yes

Is the language acceptable?

Yes

Do you have any ethical concerns with this paper?

No

Have you any concerns about statistical analyses in this paper?

No

Recommendation?

Accept as is

Comments to the Author(s)

RSOS-211146

Reviewer Comments

I was one of the reviewers on the initial submission. I was impressed by the authors' thoughtful responses to the feedback from all reviewers. The revised section on "Scientific consequences" now better represents the literature. The authors have strengthened their argument about the importance of using more diverse stimuli to answer questions about the accuracy, consensus, and origins of facial impressions. The new section, "Might evaluations based on ethnicity overshadow other types of inferences?" fully addresses my concerns about possible top-down effects of racial stereotypes – and further highlights the case for including more diverse stimuli. I think this paper is now considerably stronger and will be an important contribution to the literature.

Decision letter (RSOS-211146.R0)

Dear Dr Cook

On behalf of the Editors, we are pleased to inform you that your Manuscript RSOS-211146 "Why is the literature on first impressions so focused on White faces?" has been accepted for publication in Royal Society Open Science subject to minor revision in accordance with the referees' reports. Please find the referees' comments along with any feedback from the Editors below my signature.

Please submit your revised manuscript and required files (see below) no later than 7 days from today's (ie 25-Aug-2021) date. Note: the ScholarOne system will 'lock' if submission of the revision is attempted 7 or more days after the deadline. If you do not think you will be able to meet this deadline please contact the editorial office immediately.

Please note article processing charges apply to papers accepted for publication in Royal Society Open Science (<https://royalsocietypublishing.org/rsos/charges>). Charges will also apply to papers transferred to the journal from other Royal Society Publishing journals, as well as papers submitted as part of our collaboration with the Royal Society of Chemistry

(<https://royalsocietypublishing.org/rsos/chemistry>). Fee waivers are available but must be requested when you submit your revision (<https://royalsocietypublishing.org/rsos/waivers>).

on behalf of Dr Giorgia Silani (Associate Editor) and Essi Viding (Subject Editor)
openscience@royalsociety.org

Associate Editor Comments to Author (Dr Giorgia Silani):
Associate Editor

Comments to the Author:

We have now received the comments of the three anonymous reviewers and they all agreed that the paper has improved and is suitable for publication. Few minor suggestions are still pending. I would recommend the author to address them in a revised version of the paper. If satisfactory, I will not send the version to the reviewers anymore.

Reviewer comments to Author:

Reviewer: 2

Comments to the Author(s)

The authors sufficiently addressed my points, including citing proper previous papers and careful rephrasing of their points. I believe these changes, combined with changes suggested by other reviewers, strengthened the submitted work. I have no remaining issue other than a few minor points:

1. "We also note that certain types of character (e.g., heroes and villains, jocks and geeks) are also prone to stereotypical depiction in film, TV, comics, and storybooks. The activation of these character stereotypes may afford a range of attributions about courage, trustworthiness, and academic and sporting ability."

Perhaps the empirical studies of cultural learning of impression is relevant here as a reference, in which stereotypical depiction of a character affects people's impressions of a group of individuals (e.g., Black individuals; e.g., Weisbuch et al. 2009) including the authors' own work.

Weisbuch, M., Pauker, K., & Ambady, N. (2009). The subtle transmission of race bias via televised nonverbal behavior. *Science*, 326, 1711–1714

2. "More generally, there has been lack of explicit discussion about the relationship a between group stereotypes and other types of first impression." (Conclusion)

Perhaps not the main point here, but to avoid confusion it may be useful to mention that there has been work investigating the relationship between group stereotypes and first impressions. One example is work investigating the perceptual origin of gender/racial face-ism (group stereotypes at the level of faces). For example, different racial groups are often associated with different perceptions [44,45], which are closely emotional perceptions. Some of these associations were found even at the objective visual facial properties (e.g., Zebrowitz et al. (2010) found that White faces resembled angry faces than did East Asian and Black faces; female faces were similar

to a surprised face than were male faces. For the category vs. emotion resemblance analysis of the faces, they used connectionist models, not human judges) These findings suggest that perhaps at least partially people are intrinsically biased to form face-based trait impressions of different social categories in a somewhat fixed, different ways. Of course, we should be extra careful when interpreting this line of work, but it is worthy of noting that there are visual properties that lead to category-based face-ism, even without any conceptual stereotypes. I noticed other reviewers raised a similar point.

Also related to said "explicit discussion" is work on the relationship between racial stereotypes and first face impressions. For example, Xie et al. (accepted) found that individual perceivers' different ways of forming impressions of racially diverse faces are related to their self-reported race-related stereotypes, as assessed by second-order similarity analysis.

Both papers are a great example of scientific endeavor examining the relationship a between group stereotypes and other types of first impression (although of course they do not justify any racist narratives or atrocities).

Xie, S. Y., Flake, J. K., Stolier, R. M., Freeman, J. B., & Hehman, E. (conditionally accepted). Facial impressions are predicted by the structure of group stereotypes. *Psychological Science*.
Zebrowitz, L. A., Kikuchi, M., & Fellous, J. M. (2010). Facial resemblance to emotions: group differences, impression effects, and race stereotypes. *Journal of Personality and Social Psychology*, 98(2), 175–189.

3. "Every stimulus face used in the study described by Oh et al. (2019) was White." (Authors' response to R2 #2)

Oh et al. (2019) used racially diverse stimuli to validate their models (Study 3B), which is explicitly stated: "50 photos of eight self-identified East Asian (four females), 8 West Asian (three females), 12 Black (five females), and 22 White actors (13 females) between the ages of 19–37 were used (Supplemental Figure S8 in the online supplemental material). Non-White faces comprised 56% of all faces."

Reviewer: 1

Comments to the Author(s)

The authors' reply and revision of the manuscript are satisfactory. I only have two further comments.

emotional expressions

In their response to my previous review, the authors write that they "agree that trait evaluations based on the presence of perceived emotion likely differ from those attributable to group or character stereotypes. We now acknowledge this in the revision (see the final paragraph in the section entitled "Does the use of diverse faces introduce a confound?").

The only reference to emotion I could find in that section is in the very last sentence: "If these effects were removed from the literature, together with those effects driven by the (mis)perception of facial emotion [57, 58], we wonder: what would be left?"

I seriously doubt that the average physical resemblance of certain facial features (like those characteristic of male and female faces) with emotional facial expressions can ever be "removed" in the stimuli used so far. Did the authors mean to say that previous research reporting this association only used white faces, and that the association might disappear if a more ethnically diverse set of faces was analysed? If so, maybe they could make this thought more explicit?

implications of showing White faces only

I agree with the other reviewers that it is often advisable to reduce the dimensionality of the stimulus set, at least during initial stages of investigation, to conduct rigorous research. Of course, doing so comes to the expense of ecological validity, and one can ask how much of the results emerging from this way of doing science matter in “real” life. But the same can be said about most other aspects of experimental research, especially the kind being conducted by showing pre-recorded stimuli on computer screens in psychological laboratories. Calls to study social cognition using a more interactive ‘second-person’ approach have been around for years (Redcay and Schilbach, *Nature Reviews Neuroscience*, 2019), and corresponding efforts to fundamentally change how research in social psychology and neuroscience is conducted are underway.

Although commendable, the authors’ invitation to increase the ethnic diversity of face stimuli used in research seems far from sufficient to overcome the fundamental limitation (which is also its main strength) inherent to the scientific method, i.e. its need to isolate specific aspects of a complex phenomenon in order to tightly control them and to rule out as much as possible alternative explanations.

I also don’t think that the risk of “overshadowing” the focus of a study by introducing ethnic diversity in the stimulus set can be easily dismissed by using a blocked or between-subjects design – or at least this is an empirical question, i.e. one should first run such a study and then verify that there is no effect of block or group. It is true that “interleaving male and female faces may also render sexually dimorphic cues salient”, and that this might influence participants’ responses. Arguably, when seeing faces out of context, ‘race’ is a much more charged concept than ‘gender’, and therefore greater “overshadowing” effects can be expected to come from race than gender.

Based on these reflections, I would recommend that the authors tone down some of their statements, especially in the abstract. E.g. the sentence “the focus on White faces has undermined scientific efforts to understand first impressions from faces and [...] has served to reinforce socially regressive ideas about ‘race’”. I think it is somewhat of an exaggeration to say that the focus on White faces has undermined the understanding of first impressions, although I agree that most results obtained so far might not apply to faces from other ethnicities. I also remain sceptical (like Reviewer 3) about the claim that the focus on White faces has reinforced ideas about race, but I appreciate the authors’ arguments and accept their view on this.

Reviewer: 3

Comments to the Author(s)

RSOS-211146

Reviewer Comments

I was one of the reviewers on the initial submission. I was impressed by the authors' thoughtful responses to the feedback from all reviewers. The revised section on “Scientific consequences” now better represents the literature. The authors have strengthened their argument about the importance of using more diverse stimuli to answer questions about the accuracy, consensus, and origins of facial impressions. The new section, “Might evaluations based on ethnicity overshadow other types of inferences?” fully addresses my concerns about possible top-down effects of racial stereotypes – and further highlights the case for including more diverse stimuli. I think this paper is now considerably stronger and will be an important contribution to the literature.

===PREPARING YOUR MANUSCRIPT===

===PREPARING YOUR REVISION IN SCHOLARONE===

- An individual file of each figure (EPS or print-quality PDF preferred [either format should be produced directly from original creation package], or original software format).
- An editable file of each table (.doc, .docx, .xls, .xlsx, or .csv).
- An editable file of all figure and table captions.

- Any electronic supplementary material (ESM).
- If you are requesting a discretionary waiver for the article processing charge, the waiver form must be included at this step.
- If you are providing image files for potential cover images, please upload these at this step, and inform the editorial office you have done so. You must hold the copyright to any image provided.
- A copy of your point-by-point response to referees and Editors. This will expedite the preparation of your proof.

- Ensure that your data access statement meets the requirements at <https://royalsociety.org/journals/authors/author-guidelines/#data>. You should ensure that you cite the dataset in your reference list. If you have deposited data etc in the Dryad repository, please only include the 'For publication' link at this stage. You should remove the 'For review' link.
- If you are requesting an article processing charge waiver, you must select the relevant waiver option (if requesting a discretionary waiver, the form should have been uploaded at Step 3 'File upload' above).
- If you have uploaded ESM files, please ensure you follow the guidance at <https://royalsociety.org/journals/authors/author-guidelines/#supplementary-material> to include a suitable title and informative caption. An example of appropriate titling and captioning may be found at https://figshare.com/articles/Table_S2_from_Is_there_a_trade-off_between_peak_performance_and_performance_breadth_across_temperatures_for_aerobic_scope_in_teleost_fishes_/3843624.

Author's Response to Decision Letter for (RSOS-211146.R0)

See Appendix B.

Decision letter (RSOS-211146.R1)

Dear Dr Cook,

I am pleased to inform you that your manuscript entitled "Why is the literature on first impressions so focused on White faces?" is now accepted for publication in Royal Society Open Science.

on behalf of Dr Giorgia Silani (Associate Editor) and Essi Viding (Subject Editor)
openscience@royalsociety.org

Appendix A

Why is the literature on first impressions so focused on White faces?

Response to Reviewers

Comments to the authors: Reviewer #1

Both authors are experts on the topic of first impressions from faces, the manuscript is interesting and very well written, and I agree with (most of) its content. This being said, I would like to invite the authors to discuss a potential point of concern with their proposition of increasing the amount and the number of differences between facial stimuli in studies on first impressions.

We are grateful to the reviewer for their constructive suggestions.

The authors focus on the fact that stereotypes can influence first impressions of faces, so to say in a “top-down” way. From this perspective, the sight of a face activates numerous stereotypes (e.g., of gender), even when only white faces are used. Including more racially diverse faces therefore should not, according to the authors, introduce anything that is qualitatively different. On the contrary, they argue that it should enrich the field of first impressions studies.

However, a point that the authors seem not to have addressed is that associations between facial features and specific mental representations may sometimes operate “bottom-up”, due to morphological resemblance. For example, facial expressions of anger show more visual resemblance with the morphological features of average male faces with neutral expressions – typically characterised by prominent eyebrows, thinner lips, and a more squared jaw – than to the morphological features of average female faces (Said et al., 2009; Zebrowitz et al., 2010).

We agree that trait evaluations based on the presence of perceived emotion likely differ from those attributable to group or character stereotypes. We now acknowledge this in the revision (see the final paragraph in the section entitled “Does the use of diverse faces introduce a confound?”).

In our view, the terms “bottom-up” and “top-down” are not well-used in this area. For example, although inferences about emotion have been described as “bottom-up” they involve a great deal of interpretation and perceptual inference. When a participant encounters an unknown face with an ambiguous expression, they need to infer whether the stimulus presented depicts a person with an unusual face shape expressing no emotion or a statistically more likely face shape expressing a subtle emotion.

This begs the question, if the visual resemblance between faces at rest and specific emotional facial expressions also varies between ethnic groups (either in absolute terms, or in the context of the observer’s own ethnic background/familiarity). Assuming that such an association between ethnic background and facial expressions exists, would we not want to control for it? Neutral and emotional facial expressions result in different brain responses after just 100ms (Vuilleumier & Pourtois, 2007).

If faces from diverse ethnic backgrounds present different degrees of visible resemblance with emotional facial expressions, then striving for greater ethnic stimulus variability might induce unwanted “bottom-up” effects on behavioural and brain responses, which are distinct (independent or correlated, we do not know yet) from the effects due to stereotype activation discussed by Cook and Over.

It is known that White participants sometimes perceive Black faces to be more angry than White faces. Far from being a problem for the field, however, this is part of the phenomenon of interest. Differences in the perceived anger of Black and White faces by Black and White participants, and the potential implications for first impressions, is surely a very interesting avenue for future research.

The presence of greater diversity in stimulus sets may produce some complex results: There may be evidence of contextual variability and individual differences, and some feature-trait relationships may not generalise to all face types. However, this is not a problem with the approach *per se*. Rather, this reflects the reality of the phenomenon. We have made this point in the revised Conclusion.

Early event related potentials (ERPs) have also been shown to be sensitive to aspects of visual stimuli that typically lie outside researchers' interests, such as the power in different spatial frequency bands (Delplanque et al., 2007). Careful control of low-level stimulus features (luminance, spatial frequencies) is thus crucial for well-executed ERP studies, as well as for other methods, such as continuous flash suppression (CFS; Gray et al., 2013; Yang & Blake, 2012). While methodological constraints should not prevent us from increasing ethnic diversity in face stimulus sets, the challenges associated with it should nevertheless be acknowledged.

We agree with the reviewer that there may be some cases where, in order to maintain experimental control, researchers need to limit their stimuli to a single ethnicity. However, there is no theoretically persuasive reason to limit the faces studied to White faces specifically. Low level stimulus features could be equally controlled for by using exclusively Black faces, for example. Alternatively, White, Black, and Asian trials could be blocked, or used in a between-subjects design. We now make this point more clearly (e.g., see the section entitled "Might evaluations based on ethnicity overshadow other types of inference?").

In summary, the current manuscript makes a convincing claim for the need to include more ethnically diverse stimuli in studies investigating first impressions from faces. The authors however might want to also address that greater stimulus diversity bears the risk of complicating the task of acquiring meaningful measures of brain activity, such as EEG/ERPs, as well as using faces in some paradigms like CFS.

In the revision, we have sought to give more consideration to the potential methodological challenges associated with incorporating greater stimulus diversity (e.g., see the section entitled "Might evaluations based on ethnicity overshadow other types of inference?").

The first 2 sentences in the abstract and introduction are identical. Consider changing slightly.

We have amended the first sentence of the abstract along the lines suggested.

Comments to the authors: Reviewer #2

In the current review, the authors review the problematic tendency in the literature of the intuitive trait judgments of strangers from facial appearances ("first impressions") to use only white faces -- often all male and young. They review the rationale for this tendency described in the literature, conclude that these rationale does not withstand scrutiny, and posits including non-white faces in the investigation of the issue should be the norm, not an exception. I read the manuscript with great interest. While I agree with the general point of the paper, I do have minor but serious concerns regarding specific points found in the paper. I'd like to suggest revisions so that the paper makes its point more clear.

We are grateful to the reviewer for their constructive suggestions.

1. Overall, I am under an impression that the review paints a rather pigeonholed picture of the field, which arguably is unfair to many researchers that care about the issue of face diversity. I understand that the pigeonholing rhetoric is an effective way to convey the main idea of the paper, and I agree that in some individual empirical papers cited here, the reasons behind using only White faces are unjustifiable, for the very reasons the authors state. However, as the field began to care more about the issue of face diversity in the face sample, recent papers now use more comprehensive, diverse set of faces (as partially described in "Positive Signs" by the authors). It should be noted that these are often the same research teams (i.e., teams led by the same authors) that once used only white faces in their older papers. So, it is clear that researchers, our field as a whole, now care about the diversity in face samples.

It is absolutely not our intention to criticise particular research groups. Indeed, as we acknowledge, we have previously been guilty of using exclusively White faces in our own research (see the start of the section entitled "Why is there a lack of facial diversity in the first impressions literature?").

We suspect that authors are making different assumptions about first impressions based on perceived ethnicity. However, because these issues are rarely discussed in print, these different assumptions remain implicit. Several authors feature prominently in our review because they are more explicit about their reasons for omitting faces of colour. As we now highlight more clearly, this is actually very helpful as it provides the basis for a constructive discussion (see the section entitled "Positive developments" as well as the Conclusion).

Throughout the revision, we have been careful to avoid a rhetorical tone.

2. Speaking of old papers, I'd like to take Oosterhof & Todorov (2008) for example. This is a paper published 13 years ago, and is admittedly one of the first papers that investigated the issue of the structure of social evaluation of others' faces (and its underlying perceptual basis). This in my opinion is already a daunting task, and at that point it makes sense at least to me to keep one's stimuli homogeneous, racially and otherwise. You got to start from somewhere.

The suggestion that it was necessary to focus on White faces in order to render the problem tractable is not persuasive. The purported aim of this work was to reveal the structure of social evaluations of others' faces. How can a solution that ignores the role of race and ethnicity be deemed credible? Perceived ethnicity remains one of the most influential cues in day-to-day social evaluations in Western society. Furthermore, the majority of the earth's population do not identify as White.

We agree that it is only possible to achieve so much in a single study. However, Oosterhof and Todorov (2008) do not explicitly limit their conclusions to first impressions of White faces. Rather they talk about the functional basis of face evaluation in general. Where researchers choose to focus exclusively on perceptions of White faces, it is important they limit their conclusions accordingly.

While the reviewer refers to Oosterhof & Todorov (2008) as an "old paper", this work has been cited over 750 times since 2018 (according to Google Scholar). The interpretation of this work is very much a contemporary issue.

In the light of this, it is equally important to emphasize how the field reacted to the paper (and to similar papers that followed the paper). Naturally, the 2008 paper facilitated future research to look into the variabilities in the structure of face evaluation across face categories – e.g., races, age groups, sexes:

- races: Sutherland et al. (2018) built separate data-driven evaluation models of East Asian and Caucasian male/female faces recruiting East Asian and Caucasian participants

This study was cited in the original manuscript and is cited in the revision as a positive development.

- age groups: Collova et al. (2019) found for children's faces, unlike the original 2d model (Oosterhof & Todorov, 2008), two summary dimensions of "valence" and "shyness" implying that the face evaluation structure may be driven by social goals in the perceiver's mind (e.g., interactions with the a grown-up vs. child).

Every stimulus face used in the study described by Collova et al. (2019) was White. All the participants in this study were White. The authors cite potential concerns about the other-race effect as justification for their use of White faces and White raters.

- sexes: Oh et al. (2019) extracted separate summary dimensions of male and female faces via separate PCAs/correlations of the face-based trait ratings, and built separate data-driven trait models of male and female faces.

Every stimulus face used in the study described by Oh et al. (2019) was White. Although no details about participant ethnicity are provided, we presume they were likely from diverse ethnic backgrounds as they were recruited through MTurk. The authors offer no explanation for their use of White face stimuli or their choice of participants.

These later efforts highlight that we as a field are not stuck in a regressive mindset of using only one kind of (i.e., white) faces. The "Positive Signs" section definitely echoes the same sentiment and is great for a balanced discussion, but I think how expansive the change in the field is right now, and how seriously people are considering the issue could be better expressed.

We agree with the reviewer that there are positive signs and that some research groups are taking this issue seriously. We have sought to reference a greater number of these studies in the revision (see Section on Positive developments). We're very happy to reference additional positive examples if the reviewer feels important references are missing.

We strongly disagree, however, that this is a historical problem that has now been fixed. In the revision, we list many examples of studies published within the last 3 years that use all-White faces to evidence the fact that this is still an ongoing issue (see final paragraph of the Introduction).

3. p.12 "particularly good example is a recent study described by Jones and colleagues ... the authors found that cross-cultural differences have been under-estimated in the extant literature."

I concur that Jones et al. (2021) is an admirable project with much value. I am glad that the study included Asian, Black, and Latino faces in their stimulus set. However, I have to say this particular statement by the authors reflects an incorrect reading of the paper. Jones et al. found highly similar structure across all regions when they applied a PCA to different

world regions (the same method used by the Oosterhof and Todorov's original 2008 paper) – arguably except for the Eastern Europe (see Table 3). The previously "underestimated" cross-cultural differences emerged only due to a different *method* applied to the data between the paper and the original 2008 paper. (i.e., PCA vs. factor analysis), not different *world regions*. Moreover, the cross-regional differences may be inflated due to a methodological artifact, a recent paper from the Todorov's lab (Todorov & Oh 2021) argues, and when the artifact is adjusted, there appears to be virtually no cross-regional differences in the data.

I would like to emphasize this: It is absolutely possible to celebrate the fact that the 2021 paper employed racially diverse face sets, while admitting that there was little cross-regional difference found in the Valence-Power face evaluation structure (after all, you never know what you'll find out until you actually test your ideas). On the other hand, weighted indexes of Valence and Power of individual faces (not the Valence-Power structure per se) may be able to reveal some cross-regional differences in people's impressions of racially diverse faces. This was revealed using the same international, Jones et al. data (Todorov & Oh 2021; see Section 1.3 and Figure 6). So there may be indeed some cross-regional variabilities (induced by the racially diverse faces) there.

We apologise if our description of these results was imprecise. As the reviewer alludes to, we cited this study primarily for its methods, rather than the findings. We have amended our description of this study to make this clearer (see section entitled: "Positive developments").

4. There were several statements in the paper that lack citations that can benefit from including the exact sources.

We agree with the reviewer and have added further references throughout.

"in reality, the divide between "black" and "white" faces owes at least as much to cultural factors as to biology: two people, one who identifies as black and one who identifies as white, could easily share more genes than two people who identify as black." : I really want this to be true. However, my knowledge (or acceptance thereof) should be based on data, not my feelings. Thus, I'm keen to see a citation on this.

We have provided references for this in the revision.

"Moreover, not everyone shows other-race effects. People are thought to develop expertise for the types of faces to which they are exposed. Where first impressions research conducted in diverse urban centres (e.g. New York, London), it seems unjustified to assume that local participants lack perceptual expertise for faces of colour." Are these sentences a conjecture or empirical truth? The first sentence ("not everyone shows other-race effects") is almost certainly true given the wide individual variabilities in face processing, but it needs a citation.

We have provided references for this in the revision.

Expertise in face categories seems to require so much more than casual encounters with other categories. I'm happy to be corrected, and I'd like to see a citation.

As we now highlight in the revision (see section entitled: "Do participants lack the perceptual expertise necessary to evaluate diverse faces?"), face recognition expertise is thought to depend on individuation experience - encounters that require particular faces to be identified. Life in a diverse urban centre affords much of this experience

(e.g., attending school in London typically necessitates the individuation of class-mates and teachers from a wide range of ethnic backgrounds).

It is easy enough to identify (and exclude) participants who are unable to provide reliable ratings. For example, faces could be rated twice and the consistency of their ratings identified. We have added this suggestion in the revision within the same section.

5. As you may be already aware, it's much preferable to write "Black" and "White" with the capital B and W's (vs. b's and w's) for cultural/historical reasons. Use of Black vs. black (and White vs white) seems inconsistent in the paper (e.g., on p.6 there are many "Black faces"s). I suggest "Black" and "White" throughout.

We are grateful for this thoughtful observation. We have amended our writing accordingly.

6. In Figures 1, there's a letter "d" inside the panel a.

Thank you for spotting this – the figure has been corrected.

Comments to the authors: Reviewer #3

This paper gives some consideration to the lack of racial diversity in the first impressions literature. The authors assert that the field has traditionally focused on White raters evaluating White faces, often without justification, and argue for the inclusion of more racially diverse stimuli. They describe some common justifications for using ethnically homogeneous stimuli—for instance, the concern that inferences from ‘ethnicity cues’ and ‘morphological face cues’ are qualitatively different, or that diverse faces distort findings by introducing noise (e.g., stemming from a lack of perceptual expertise viewing other-race faces). The authors argue that these justifications do not hold up to scrutiny, for example observing that many studies use computer-generated stimuli with which participants have little perceptual expertise—yet are still considered useful.

Overall, the paper is well-written, and I agree with the authors’ thesis that the lack of facial diversity is a problem. Although I found some of their arguments more compelling than others, I agree with the message that emphasizing white faces means missed opportunities for understanding how impressions are formed across group boundaries. Below I have some comments and questions that I hope to see the authors address.

We are grateful to the reviewer for their constructive suggestions.

1) The main issue addressed in this review paper is important. Although I agree with the authors’ thesis, I very much struggled with the central premise that the focus on white faces has “served to reinforce socially regressive ideas about ‘race’”. The authors repeatedly make this argument throughout the paper, for example on page 9, where the authors speculate that “the focus on white faces may have also had damaging social consequences. It may have inadvertently reinforced the regressive idea that there are discrete racial categories each with a particular genetic profile.” This argument seems to hinge on the assumption that the public either reads scientific papers about impression formation or pays close attention to the stimuli used in these studies. Or that scientific discourse has otherwise trickled into public consciousness. I have some doubts about whether that’s the case, and how much the use of stimuli in face perception research might really be contributing to the broader perception of race as categories. If the authors wish to maintain this argument, I’d be interested in empirical evidence supporting their position. But I think the authors can make the same arguments without including this issue at all, and it would not detract from the

value of their paper.

We have clarified our argument here (see final paragraph of the section entitled “Social consequences”). While this objection took us by surprise, we are grateful to the reviewer for raising this issue. We think our argument has been strengthened as a result.

We believe that there is a trickle-down effect from the scientific literature into the mainstream media and public consciousness. This is particularly the case with a “media-friendly” topic like first impressions from faces. In the revision we provide references to pieces in the mainstream media, all of which depict White faces.

Here’s a link to one of the cited articles published in 2017 in the Independent – a national newspaper published in the UK. All of the images depicted are taken from the study materials.

<https://www.independent.co.uk/life-style/here-s-how-people-judge-you-based-your-face-a7988901.html>

2) At times, I felt the authors misrepresented some of the literature which hampered my overall enthusiasm of this work. For example, the last paragraph beginning on page 8 implies that viewing first impressions as a “genetic adaptation” might lead to an “implausible, and deeply offensive” perspective that “first impressions of Black men as untrustworthy and aggressive are accurate”. I wonder if the authors have conflated adaptive and accurate here. Ecological theories of face perception assert that the ability to quickly extract social information from faces is adaptive (Zebrowitz, 2006), not necessarily accurate. That is, from a “risk management” perspective, it may be more adaptive to (incorrectly) judge an outgroup member as untrustworthy than to (incorrectly) judge an outgroup member as trustworthy (Zebrowitz, Fellous, Mignault & Adreoletti, 2003; Zebrowitz, Kikuchi, & Fellous, 2010). As such, facial impressions may have a functional basis even if they are biased or inaccurate.

It also seems like an odd segue from the previous paragraph, where the authors talk about the accuracy of trustworthiness judgments, but then jump to negative racial stereotypes about trustworthiness. Given that trustworthy might mean different things in different contexts (Wilson & Rule, 2017), I believe this paragraph could be improved by simply stating that humans aren’t very accurate at judging trustworthiness, and that there’s an abundance of literature to demonstrate that (see Todorov, Olivola, Dotsch, & Mende-Siedlecki, 2015).

In deference to the strong feelings of the reviewer, we have reworked this section (entitled “Scientific consequences”). We have sought to improve the flow of this section and we now discuss the topic of accuracy as important and interesting in its own right. We also more clearly cite the work of Todorov who has argued that first impressions are often inaccurate. Throughout the revision, we have been careful to avoid a rhetorical tone.

Part of the interest in the accuracy of first impressions certainly stems from the origins question. If first impressions are accurate, then this affords a simple narrative about the advantage conveyed on our ancestors, and the means by which the mechanism responsible could have become a genetic adaptation. In the revision we have cited a prominent nativist account that explicitly draws on this logic. However, we are careful to avoid any suggestion that all nativist accounts make this assertion. Whether inferences can be adaptive without being accurate is a controversial question, and interrogating this suggestion is beyond the scope of this manuscript.

We believe that the questions to which we allude at the end of this section (Do humans possess innate representations about both White and non-White faces? Do people

who identify as White, Black, and Asian possess similar innate representations? Do racial stereotypes have an innate basis?) have not been clearly addressed by proponents of the nativist view. We feel that this has escaped scrutiny because of the prevailing focus on White faces and White participants. It therefore seems an entirely appropriate way to end this section on how the focus on White faces has at times hindered our understanding of first impressions.

In our view, these questions apply equally to ecological theories that appeal to innate ingroups and outgroups. Is the implication that our genetic make-up causes us to equate “different race” with “outgroup”? Any suggestion that people who identify as White and Black are innately endowed with different cognitive representations is hard to reconcile with the lack of genes coding for “race” and evidence that racial categories are social constructions.

3) I was not convinced by the argument that just because White faces also elicit group stereotypes like “jock” and “geek” (end of page 6), that group stereotypes “within” the same “race” are qualitatively the same as group stereotypes “between races”.

This is a straw-man characterisation of our argument: Jocks and Geeks were only one of the examples we gave. What about people who identify as White, but activate Jewish or Italian American stereotypes? What about homophobic stereotypes? What about the group stereotypes activated by men and women?

We do not seek to equate the “jock” and “geek” stereotypes with the stereotypes associated with different ethnic groups. Clearly, the latter lead to more profound social problems. Moreover, Jocks and Geeks are not typically regarded as social groups. In the revision we have therefore made a distinction between “character stereotypes” and “group stereotypes”.

However, there are some similarities: most importantly in the present context, these stereotypes link expectations about facial appearance and expected traits. Various cultural sources (e.g., TV, film, movies, comics, storybooks) provide stereotypical depictions of heroes and villains, geeks and jocks. The means by which we acquire these stereotypes might be similar to the cultural origins of racial stereotypes (which are also prevalent in TV, film and other forms of media).

While categorical thinking may be inevitable, some researchers may want to randomize out top-down categorical effects (e.g., stereotyping) as much as they can. For example, while it’s possible that a group of white faces will elicit stereotypes like “plant lover”, “geeks”, “jocks”, “hipsters”, etc., it’s far more likely that the stimulus set will randomly sample from a universe of (white) people stereotypes. However, if you use a stimulus set comprising white, black, and Korean faces, you are not randomly sampling from a universe of racial groups but emphasizing the specific racial categories in the stimulus set. It seems likely that emphasizing these racial categories would induce stronger top-down effects of racial stereotypes. Researchers who want to minimize top-down categorical effects as much as possible would have reason to avoid using mixed-race stimuli in this way, or use alternative data collection designs (i.e., blocking by race) to avoid this issue.

We thank the reviewer for this suggestion. This is an interesting possibility that we consider in detail in the revision (see the new section: “Might evaluations based on ethnicity overshadow other types of inference?”). As the reviewer alludes to, any potential issue is easily overcome by the use of blocked or between-subjects designs.

4) Somewhat related to the above comment, I think the strongest argument for including racially diverse stimuli is that it’s important to test whether findings generalize to people from

other racial groups. Any model that's developed on white faces, if intended to be universally applicable, should fit well for other groups. The authors do mention that an influential model of face perception does not readily generalize to other cultures (Jones et al., 2021), and I expected this to be framed as an important demonstration of the value of using racially diverse stimuli. Although this is just my opinion, I consider this to be the most important point, and thought it could be emphasized more.

We agree with the reviewer that this is an important point. We have sought to highlight this issue further in the revision. For example, we make this point in the new section "Might evaluations based on ethnicity overshadow other types of inference?" and in the Conclusion.

5) I appreciate that the authors have cited broadly to demonstrate the overreliance on white faces in the first impressions literature. However, a growing body of work does use more racially diverse stimuli, some of which the authors cite (Jones et al., 2021) and some they do not. Some of these papers speak directly to the claims made by the authors, and rather than speculate, they could just cite the existing evidence.

For example, the authors can back up their speculation on page 10 that "Greater diversity in stimulus sets is therefore likely to reveal more heterogenous first impressions" by citing Xie, Flake, and Hehman (2019), which uses a large, racially diverse stimulus set to examine the variability in facial impressions across different race and gender groups. Regardless of target race, "perceiver idiosyncrasies" contribute more variance to face impressions (~25%) than do "shared consensus" (~15%), providing empirical support for the claim that any consensus in facial judgments might be overstated when participants are evaluating racially diverse faces.

A notable omission is a recent paper by Stolier, Hehman, Keller, Walker, and Freeman (2018), which finds that perceivers' own lay theories of personality (trait space) map onto the structure of their facial impressions (face space) even when participants are only evaluating White male stimuli. This speaks directly to the authors' argument that trait inferences from facial cues may be shaped by participants' experiences, and is also relevant to the argument that top-down processing occurs even in racially homogeneous stimuli sets.

We thank the reviewer for these useful suggestions. These findings are cited in the revision.

6) While this is a paper about face perception, the first sentence is too strong and somewhat incorrect, and I interpreted it as a typo. The authors state, "When we first encounter a stranger, we spontaneously attribute to them a wide variety of character traits based solely on their facial appearance." As people use other cues to form an impression (e.g., body, clothes, hairstyle), I suggest removing "solely" from this sentence.

We have amended this wording accordingly.

7) When discussing the 'other-race effect', the authors talk about it entirely from the perceptual expertise account but do not mention motivational accounts of other-race effects. Can the authors consider the relevance of this motivational component and how it might impact their conclusions? From this perspective, familiarity with the outgroups likely wouldn't matter for impacting trait impressions from faces (see Young, Hugenberg, Bernstein, & Sacco, 2012).

There is broad consensus that perceptual expertise (or a lack thereof) is a significant determinant of the other-race effect. In the eyes of many authors, this is the sole cause of the effect. Some authors have argued that this effect also has a motivational

component: when viewing outgroup faces people may be less motivated to individuate outgroup faces than ingroup faces (e.g., Young et al., 2012). However, the evidence for this view is less consistent and this possibility remains quite controversial.

Importantly, where authors have cited the other-race effect as a rationale for excluding faces of colour, they have alluded to perceptual problems (not motivational differences). This concern is therefore the focus of our discussion.

Obviously, we cannot rule out the fact that people, born and raised in diverse urban centres (e.g., London), might show weak other-race effects either due to residual differences in their “daily diet of faces” or due to motivational factors. Our point is that there is every reason to suspect they have adequate perceptual ability to provide meaningful trait evaluations of diverse faces.

It is easy enough to identify (and exclude) participants who are unable to provide reliable ratings e.g., faces could be rated twice and the consistency of their ratings examined. We have added this suggestion in the revision (see section: Do participants lack the perceptual expertise necessary to evaluate diverse faces?).

8) Somewhat related to an earlier point, presenting racially heterogeneous stimuli to participants may change the way that they anchor their impressions. Is there a difference between using a racially diverse stimuli set presented in a racially homogeneous way (e.g., all black, all white, between-participants) versus presenting a racially heterogeneous presentation (50% black, 50% white)? What are some recommendations the authors can make for including diverse stimuli in studies (e.g., separate batches, mixed in equal amounts, mixed in some other way)? I wasn't sure what evidence was out there that might speak to this.

We don't think that there's a “right way” or a “wrong way” to introduce greater diversity into this literature. Instead, we see a host of interesting empirical questions.

It may well be the case that increasing the diversity present within stimulus sets changes the way people make trait evaluations (e.g., some features may be rendered more salient, while other features may appear less salient). Importantly, however, this approximates more closely the situation that participants encounter outside the lab. Our societies are increasingly diverse. The faces we encounter in our daily lives are diverse, and our spontaneous trait judgements are made in that context.

The results may be more complex: There may be evidence of contextual variability and individual differences, and some feature-trait relationships may not generalise to all face types. However, this is not a problem with the approach *per se*; rather, this reflects the reality of the phenomenon.

We have sought to address these issues in the revision (in the new section “Might evaluations based on ethnicity overshadow other types of inference?” and in the Conclusion).

9) I appreciated that the authors pointed readers to several databases with more racially diverse stimuli, and would like to suggest a few more that the authors might be unaware of.

We thank the reviewer for suggesting these excellent additions which have been incorporated in the revision (see section “Are there logistical impediments that prevent the use of diverse stimuli?”).

Appendix B

Why is the literature on first impressions so focused on White faces?

RSOS-211146

Response to reviewers

Comments to the authors: Reviewer #1

The authors' reply and revision of the manuscript are satisfactory. I only have two further comments.

Emotional expressions. In their response to my previous review, the authors write that they “agree that trait evaluations based on the presence of perceived emotion likely differ from those attributable to group or character stereotypes. We now acknowledge this in the revision (see the final paragraph in the section entitled “Does the use of diverse faces introduce a confound?”).

The only reference to emotion I could find in that section is in the very last sentence: “If these effects were removed from the literature, together with those effects driven by the (mis)perception of facial emotion [57, 58], we wonder: what would be left?”

I seriously doubt that the average physical resemblance of certain facial features (like those characteristic of male and female faces) with emotional facial expressions can ever be “removed” in the stimuli used so far. Did the authors mean to say that previous research reporting this association only used white faces, and that the association might disappear if a more ethnically diverse set of faces was analysed? If so, maybe they could make this thought more explicit?

We apologise for any confusion here and have reworded this sentence (bottom of p6). Our point is simply that the mis-perception of emotion and culturally learned stereotypes may be the two principal sources of first impressions – if you remove effects attributable to these sources from the literature, it isn't clear what would be left.

Implications of showing White faces only. I agree with the other reviewers that it is often advisable to reduce the dimensionality of the stimulus set, at least during initial stages of investigation, to conduct rigorous research. Of course, doing so comes to the expense of ecological validity, and one can ask how much of the results emerging from this way of doing science matter in “real” life. But the same can be said about most other aspects of experimental research, especially the kind being conducted by showing pre-recorded stimuli on computer screens in psychological laboratories. Calls to study social cognition using a more interactive ‘second-person’ approach have been around for years (Redcay and Schilbach, Nature Reviews Neuroscience, 2019), and corresponding efforts to fundamentally change how research in social psychology and neuroscience is conducted are underway.

Although commendable, the authors' invitation to increase the ethnic diversity of face stimuli used in research seems far from sufficient to overcome the fundamental limitation (which is also its main strength) inherent to the scientific method, i.e. its need to isolate specific aspects of a complex phenomenon in order to tightly control them and to rule out as much as possible alternative explanations.

We accept that hard choices must sometimes be made in order to render a problem tractable, and often there is an associated loss of ecological validity. However, we do not accept that this is a fair characterisation of the prevailing focus on White faces within the first impressions literature. The issue here is more fundamental: the use of ethnically homogenous stimuli and raters profoundly distorts the phenomenon that researchers seek to understand. Cues to ethnicity are surely one of the most significant sources of impressions. The lack of diversity amongst raters and within stimulus sets has resulted in misleading conclusions about the degree of inter-rater consensus and judgement veracity.

If one wanted to understand how a car works, an analysis based solely on the wheels would yield incomplete and misleading conclusions (e.g., it would overlook the role of the engine, the nature of the fuel, breaking, steering, etc). Few would describe this narrow approach as “good science” because it renders a complex problem tractable.

I also don't think that the risk of “overshadowing” the focus of a study by introducing ethnic diversity in the stimulus set can be easily dismissed by using a blocked or between-subjects design – or at least this is an empirical question, i.e. one should first run such a study and then verify that there is no effect of block or group. It is true that “interleaving male and female faces may also render sexually dimorphic cues salient”, and that this might influence participants' responses. Arguably, when seeing faces out of context, ‘race’ is a much more charged concept than ‘gender’, and therefore greater “overshadowing” effects can be expected to come from race than gender.

As we acknowledge on p9, in certain situations, authors may have good reason to use faces of a particular ethnicity. Where this happens, however, authors must limit their conclusions accordingly; e.g., these effects were only seen with White faces encountered in an ethnically homogenous context.

Our social environments are increasingly diverse and our first impressions are formed within this context. If over-shadowing occurs, we would encourage our colleagues to see this as integral part of the phenomenon we are studying, not a problematic influence that must be controlled for. If the trait evaluations of White faces are modulated when target faces are presented within ethnically homogenous vs. ethnically diverse stimulus sets, these effects are important and need to be understood.

Based on these reflections, I would recommend that the authors tone down some of their statements, especially in the abstract. E.g. the sentence “the focus on White faces has undermined scientific efforts to understand first impressions from faces and [...] has served to reinforce socially regressive ideas about ‘race’”. I think it is somewhat of an exaggeration to say that the focus on White faces has undermined the understanding of first impressions, although I agree that most results obtained so far might not apply to faces from other ethnicities. I also remain sceptical (like Reviewer 3) about the claim that the focus on White faces has reinforced ideas

about race, but I appreciate the authors' arguments and accept their view on this.

If one accepts: i) that greater diversity will increase evidence of cultural differences, ii) that greater diversity will undermine evidence that first impressions are accurate, and iii) that greater diversity may modulate the first impressions made about White faces (i.e., "over-shadowing"), then the conclusion that "the lack of diversity has hindered scientific efforts to understand first impressions" seems inescapable.

The socially regressive influence of the prevailing research practice is easily over-looked, particularly by researchers who themselves identify as White. As we acknowledge in the paper, we were slow to recognise the problem ourselves. At least three issues can be delineated: 1) Generalising conclusions from the study of White faces and White participants to all faces and all participants without qualification reinforces the idea that White faces are "standard". 2) Using only White face stimuli implies that understanding first impressions about White faces is somehow more important. 3) Failure to acknowledge that "racial" categories are largely social constructions reinforces old-fashioned and regressive ideas about race. We have sought to clarify our argument in the revision (p13).

Comments to the authors: Reviewer #2

The authors sufficiently addressed my points, including citing proper previous papers and careful rephrasing of their points. I believe these changes, combined with changes suggested by other reviewers, strengthened the submitted work. I have no remaining issue other than a few minor points:

1. "We also note that certain types of character (e.g., heroes and villains, jocks and geeks) are also prone to stereotypical depiction in film, TV, comics, and storybooks. The activation of these character stereotypes may afford a range of attributions about courage, trustworthiness, and academic and sporting ability."

Perhaps the empirical studies of cultural learning of impression is relevant here as a reference, in which stereotypical depiction of a character affects people's impressions of a group of individuals (e.g., Black individuals; e.g., Weisbuch et al. 2009) including the authors' own work.

Weisbuch, M., Pauker, K., & Ambady, N. (2009). The subtle transmission of race bias via televised nonverbal behavior. *Science*, 326, 1711–1714

This is an excellent suggestion. We have added this reference (p4).

2. "More generally, there has been lack of explicit discussion about the relationship a between group stereotypes and other types of first impression." (Conclusion)

Perhaps not the main point here, but to avoid confusion it may be useful to mention that there has been work investigating the relationship between group stereotypes and first impressions. One example is work investigating the perceptual origin of gender/racial face-ism (group stereotypes at the level of faces). For example, different racial groups are often associated with different perceptions [44,45], which

are closely emotional perceptions. Some of these associations were found even at the objective visual facial properties (e.g., Zebrowitz et al. (2010) found that White faces resembled angry faces than did East Asian and Black faces; female faces were similar to a surprised face than were male faces. For the category vs. emotion resemblance analysis of the faces, they used connectionist models, not human judges) These findings suggest that perhaps at least partially people are intrinsically biased to form face-based trait impressions of different social categories in a somewhat fixed, different ways. Of course, we should be extra careful when interpreting this line of work, but it is worthy of noting that there are visual properties that lead to category-based face-ism, even without any conceptual stereotypes. I noticed other reviewers raised a similar point.

Also related to said "explicit discussion" is work on the relationship between racial stereotypes and first face impressions. For example, Xie et al. (accepted) found that individual perceivers' different ways of forming impressions of racially diverse faces are related to their self-reported race-related stereotypes, as assessed by second-order similarity analysis.

Both papers are a great example of scientific endeavor examining the relationship a between group stereotypes and other types of first impression (although of course they do not justify any racist narratives or atrocities).

Xie, S. Y., Flake, J. K., Stolier, R. M., Freeman, J. B., & Hehman, E. (conditionally accepted). Facial impressions are predicted by the structure of group stereotypes. *Psychological Science*.

Zebrowitz, L. A., Kikuchi, M., & Fellous, J. M. (2010). Facial resemblance to emotions: group differences, impression effects, and race stereotypes. *Journal of Personality and Social Psychology*, 98(2), 175–189.

We thank the reviewer for suggesting these references which have been added to the revision (p15).

3. "Every stimulus face used in the study described by Oh et al. (2019) was White." (Authors' response to R2 #2)

Oh et al. (2019) used racially diverse stimuli to validate their models (Study 3B), which is explicitly stated: "50 photos of eight self-identified East Asian (four females), 8 West Asian (three females), 12 Black (five females), and 22 White actors (13 females) between the ages of 19–37 were used (Supplemental Figure S8 in the online supplementary material). Non-White faces comprised 56% of all faces."

We apologise for any confusion here. There are two similar papers published by Oh and colleagues one published in *Psych Science* in 2019 and one published in *JEPG*. The online version of the *JEPG* paper appeared in 2019 but was published in a 2020 issue. As the reviewer states, Oh et al (2020, *JEPG*) use racially diverse stimuli to validate their models. However, we were referring to Oh et al (2019, *Psych Sci*) which used only White face stimuli. We have carefully checked out treatment of these article to ensure they are referenced appropriately.

Comments to the authors: Reviewer #3

I was one of the reviewers on the initial submission. I was impressed by the authors' thoughtful responses to the feedback from all reviewers. The revised section on "Scientific consequences" now better represents the literature. The authors have strengthened their argument about the importance of using more diverse stimuli to answer questions about the accuracy, consensus, and origins of facial impressions. The new section, "Might evaluations based on ethnicity overshadow other types of inferences?" fully addresses my concerns about possible top-down effects of racial stereotypes—and further highlights the case for including more diverse stimuli. I think this paper is now considerably stronger and will be an important contribution to the literature.

We thank the reviewer for their constructive suggestions.